# What If We Allocate Test-Time Compute Adaptively?

**Ahsan Bilal**[† 1]   **Muhammad Ahmed Mohsin**[† 2]   **Muhammad Umer**[2]   **Ali Subhan**[3]   **Hassan Rizwan**[4]
**Ayesha Mohsin**[5]   **Dean F. Hougen**[1]

## Abstract

Test-time compute scaling allocates inference computation uniformly, uses fixed sampling strategies, and applies verification only for reranking. In contrast, we propose a verifier-guided adaptive framework treating reasoning as iterative trajectory generation and selection. For each problem, the agent runs multiple inference iterations. In each iteration, it optionally produces a high-level plan, selects a set of reasoning tools and a compute strategy together with an exploration parameter, and then generates a candidate reasoning trajectory. A process reward model (PRM) serves as a unified control signal: within each iteration, step-level PRM scores are aggregated to guide pruning and expansion during generation, and across iterations, aggregated trajectory rewards are used to select the final response. Across datasets, our dynamic, PRM-guided approach consistently outperforms direct test-time scaling, yielding large gains on MATH-500 and several-fold improvements on harder benchmarks such as AIME24 and AMO-Bench. We characterize efficiency using theoretical FLOPs and a compute intensity metric penalizing wasted generation and tool overhead, demonstrating that verification-guided allocation concentrates computation on high-utility reasoning paths.

## 1. Introduction

The paradigm for improving LLM reasoning performance has increasingly shifted from scaling model parameters during pretraining to scaling computation at test time (Snell et al., 2025; Brown et al., 2024). This shift reframes infer-

ence as an optimization problem: given additional inference-time computation, the objective is no longer simply to generate an answer, but to decide how that computation should be allocated across reasoning steps, strategies, and alternatives to maximize solution reliability and accuracy. Current test-time compute scaling methods, however, face three fundamental limitations in addressing this optimization problem.

First, they lack control over where and how additional inference-time computation is applied: expensive reasoning procedures are typically applied uniformly across all inputs, wasting computation on easy problems while under-allocating exploration to genuinely hard instances (Brown et al., 2024).

Second, inference strategies are typically fixed a priori, for example, by pre-specifying the number of samples in Best-of-$N$ or the depth of a search procedure, rather than adapting based on intermediate reasoning quality (Rakhsha et al., 2025). As a result, the same exploration budget is applied regardless of whether early reasoning steps are clearly correct or already flawed.

Third, while several recent methods incorporate intermediate verification or scoring to guide search (e.g., via MCTS-style expansion and pruning (Lightman et al., 2024; Xie et al., 2024)), such mechanisms are typically tied to a *fixed* inference procedure and act only within a predetermined search algorithm. In contrast, existing approaches do not use verification signals to dynamically select *which* reasoning tools to invoke or *which* compute strategy to apply on a per-problem basis. As a result, verification is used only to rerank completed candidates produced by a predetermined inference procedure, rather than to influence which reasoning tools are invoked or how much computation is allocated while reasoning is still in progress.

We address this challenge in the context of *mathematical reasoning* with a *verifier-guided adaptive test-time inference framework* that treats reasoning as an iterative, trajectory-level decision process rather than a single-shot generation. Mathematical problem solving is particularly sensitive to early errors and branching choices, making it a natural testbed for adaptive allocation of test-time compute (Mirzadeh et al., 2024). While we focus on mathematics, the framework is general and can be applied to other

---

[†]Equal contribution. [1]University of Oklahoma, OK, USA [2]Stanford University, CA, USA [3]Universitat Pompeu Fabra, Barcelona, Spain [4]University of California, Riverside, CA, USA [5]National University of Sciences and Technology, Islamabad, Pakistan. Correspondence to: Ahsan Bilal <ahsan.bilal-1@ou.edu>.

*Proceedings of the 43$^{rd}$ International Conference on Machine Learning*, Seoul, South Korea. PMLR 306, 2026. Copyright 2026 by the author(s).

reasoning-intensive domains where intermediate verification signals are available. For each problem, the agent runs multiple iterations, selecting tools and compute strategies per iteration and using step-level PRM scores to guide trajectory generation and final answer selection.

**Contribution.** We make three main contributions:

1. **Trajectory-level formulation.** We cast adaptive test-time reasoning as a trajectory-level generation and selection problem, in which reasoning is produced incrementally and evaluated at each step. A process reward model (PRM) provides local correctness signals during generation, grounding trajectory construction in verified step transitions and actively steering the model toward *better reasoning paths* as they are formed. This enables online pruning to improve reasoning quality.

2. **Empirical gains over static inference strategies.** On MATH-500, adaptive inference improves accuracy from $43.8\%$ to $65.4\%$ (+21.6) for Llama-3.1-8B-Instruct and from $71.2\%$ to $81.4\%$ (+10.2) for Qwen-2.5-7B-Instruct; on AIME24, it improves from $3.3\%$ to $10.0\%$ and from $6.67\%$ to $13.3\%$, respectively, and on AMO-Bench it doubles accuracy from $2.0\%$ to $4.0\%$ for both models.

3. **Compute-efficiency analysis.** We evaluate inference-time efficiency using two complementary metrics: theoretical FLOPs $F_{\text{theo}}$, which accounts for the total computation across all model invocations, and a compute intensity score $S_{\text{CI}}$, which penalizes redundant generation and verification overhead. Together, these metrics show that performance gains arise from selectively allocating computation to high-utility reasoning trajectories, rather than uniformly increasing inference-time compute.

## 2. Methodology

**Problem Formulation.** Given a problem $x$, the agent allocates inference-time computation by generating multiple reasoning trajectories across iterations. In each iteration, a specific configuration of reasoning tools (e.g., Chain-of-Thought, self-reflection, numeric or logical verification) and a compute strategy (e.g., Best-of-$N$, beam search, or lookahead with an exploration parameter) is selected, defining how the trajectory is generated. During generation, a PRM-based verifier evaluates intermediate reasoning steps and completed trajectories, assigning higher scores to locally consistent and mathematically valid transitions. These step-level signals are aggregated to prioritize promising reasoning paths, prune low-quality continuations, and ultimately favor trajectories that maintain consistent local correctness, which empirically leads to more reliable final answers. Each iteration yields a candidate trajectory, and the final output is selected via PRM-guided trajectory selection. Although we describe the framework using a trajectory-level decision

formalism, the controller is not a learned policy. Instead, adaptivity is implemented entirely at inference time via deterministic, role-conditioned prompt templates applied to the same base LLM for planning, tool selection, compute strategy selection, verification, and answer extraction, without any pretraining, finetuning, or parameter updates as detailed in Appendix B. Finally, we analyze the compute metrics $F_{\text{theo}}$ and $S_{\text{CI}}$ to show that additional computation is selectively concentrated on high-utility reasoning trajectories rather than being uniformly scaled.

**Agent Architecture.** To operationalize this adaptive framework, we design a modular agent with four hierarchical components that progressively refine the reasoning strategy for each problem, as shown in Figure 1. The agent does not learn a policy; instead, each component operates via heuristic, prompt-conditioned decisions using the same base LLM.

1) First, a *planning agent* $\mathcal{A}_P$ generates a structured high-level plan $\pi = \mathcal{A}_P(x)$ before detailed reasoning to guide downstream tool and strategy selection and is conditioned on problem characteristics.

2) Second, the *tool selection agent* $\mathcal{A}_T$ dynamically chooses a subset of reasoning tools $\mathcal{T} = \mathcal{A}_T(x, \pi)$ from the toolkit as shown in the Tool Selector in Figure 1. Decisions in $\mathcal{A}_T$ are produced by a structured, role-specific prompt that maps $(x, \pi)$ to a tool subset under explicit selection rules as detailed in Appendix B.3. Each tool serves a distinct purpose: Chain-of-Thought (CoT) performs step-by-step problem decomposition; Self-Reflection (SR) enables iterative refinement through self-critique; general Verifier (V) checks logical consistency; Numeric Verifier (NV) validates numerical correctness; Reframer (R) reformulates ambiguous problems; and Summarizer (S) compresses long trajectories.

3) Third, the *compute selection agent* $\mathcal{A}_C$ determines both the inference strategy $c$ and its exploration parameter $m \in \mathbb{N}^+$ via $c, m = \mathcal{A}_C(x, \pi, \mathcal{T})$. Here, $m$ specifies the exploration budget associated with the chosen strategy: $N = m$ for Best-of-$N$ (BoN), beam width $k = m$ for Beam Search (BS), and lookahead depth $d = m$ for Lookahead Search (LA). The compute selector uses a structured prompt conditioned on $(x, \pi, \mathcal{T})$ to jointly select $c, m$ as detailed in Appendix B.4. These strategies trade off between exploration and exploitation: small $m$ commits early to a single trajectory, while larger $m$ explores multiple candidates to mitigate early errors at a higher compute cost. Jointly selecting $(c, m)$ enables per-problem adaptation, avoiding under-exploration on hard problems and wasted compute on easy ones. Tables 2 and 9 present comprehensive ablations across all tool-strategy-parameter combinations.

4) Finally, the *answer extraction agent* $\mathcal{A}_F$ synthesizes the final answer $y = \mathcal{A}_F(\tau)$ from the completed trajectory

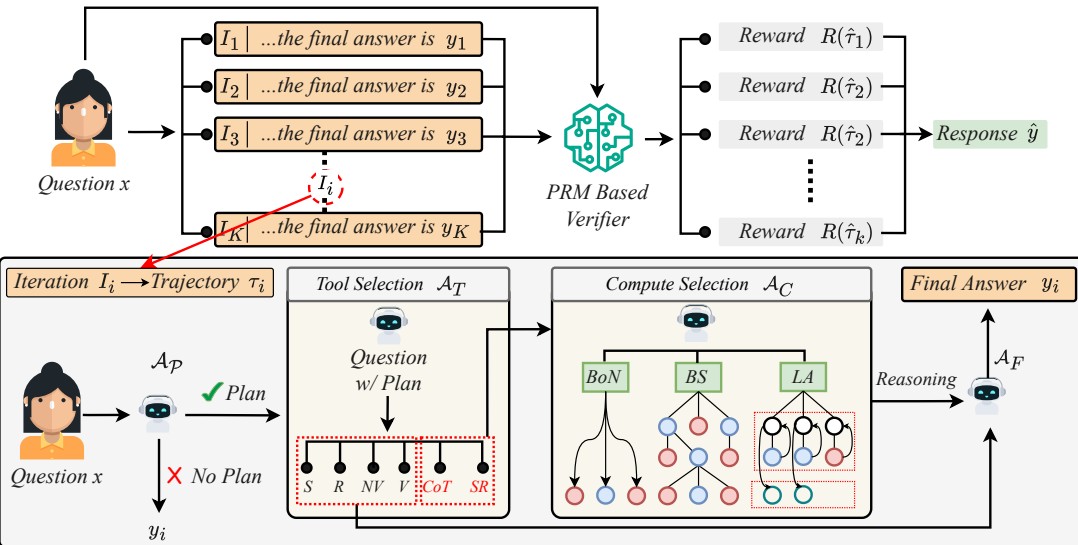

*Figure 1.* **Universal reasoning agent architecture.** The system generates $K$ candidate trajectories $\{I_1, \ldots, I_K\}$ with answers $\{y_1, \ldots, y_K\}$, scored by a PRM to produce rewards $R$ for each trajectory $\tau_i$ for selecting the best response (top). Each iteration follows a four-stage workflow (bottom): planning ($\mathcal{A}_P$), tool selection ($\mathcal{A}_T$), compute strategy selection ($\mathcal{A}_C$), and answer extraction ($\mathcal{A}_F$).

$\tau = (s_1, \ldots, s_T)$ in standardized format as detailed in Appendix B.7. This modular architecture cleanly decouples reasoning strategy selection (tools $\mathcal{T}$), exploration extent, and output formatting, enabling independent study of each component's contribution while maintaining flexible composition.

**Process Reward Models and Multi-Iteration Selection.** We use a PRM as a unified control signal for both *intra-iteration compute allocation* and *inter-iteration selection*. In all experiments, we employ a pretrained, math-specialized PRM based on **Qwen2.5-Math-PRM-7B** (Zhang et al., 2025). In our implementation, the PRM is applied as a *local step validator*: it evaluates the correctness of individual reasoning steps rather than directly predicting final-answer correctness. Verification tools, including the general Verifier and the Numeric Verifier, only produce additional reasoning steps. All such steps are evaluated uniformly by the PRM. Concretely, each tool emits step-delimited transitions $(s_{t-1} \to s_t)$ that are scored by the same PRM using identical inputs and aggregation, without access to tool identity. Thus, Verifier and Numeric Verifier differ only in the type of reasoning they generate, while correctness assessment is always performed by the PRM.

***Step segmentation and scoring.*** A reasoning trajectory is denoted $\tau = (s_1, \ldots, s_T)$, where each $s_t$ is an atomic reasoning step generated incrementally at explicit step boundaries, determined jointly by the selected reasoning tools and compute strategy. For each step transition $(s_{t-1} \to s_t)$, the PRM assigns a scalar validity score $v_t = \mathrm{PRM}(s_{t-1}, s_t) \in [0, 1]$ that evaluates local correctness while intentionally ignoring global strategy, in accordance with the step-level

prompting in Appendix B.6. Within a single trajectory (intra-iteration selection as shown at the bottom of Figure 1), PRM scores are used to guide step-level selection among candidate continuations, enabling intra-trajectory control during generation. Across iterations, i.e., inter-iteration selection as shown at the top of Figure 1, each iteration produces a completed trajectory, and PRM-based trajectory scores are used to select the final output via inter-iteration selection.

$$R(\tau) \triangleq R_{\mathrm{mean}}(\tau) = \frac{1}{T} \sum_{t=1}^{T} v_t, \qquad (1)$$

which measures aggregated local correctness rather than direct final-answer accuracy and provides a stable estimate of the overall trajectory of sound reasoning.

*1) Intra-iteration control.* Within each iteration, PRM scores are used online to guide compute allocation. Generation is step-delimited, and the PRM evaluates individual step transitions $(s_{t-1} \to s_t)$ by taking the previous and current reasoning steps as input and outputting a scalar validity score that reflects local mathematical correctness. These step-level scores are aggregated online to choose among candidate continuations (and prune low-scoring ones) before the iteration completes. Across inference strategies, intra-iteration control follows a common rule: among candidate prefixes or continuations at a given step, we retain the option with the highest aggregated PRM score.

For *Best-of-$N$*, we sample $N$ independent trajectories within iteration $\mathcal{I}_i$ and select $\hat{\tau}_i = \arg\max_{\tau \in \mathcal{S}_i} R(\tau)$. Here, the PRM is used to select the best completed trajectory within an iteration, rather than to prune partial prefixes during

generation.

For *Beam Search*, we maintain a beam set $\mathcal{B}_t$ over partial prefixes and prune candidates using the mean step validity accumulated so far, $R(\tau_{1:t}) = \frac{1}{t}\sum_{u=1}^{t} v_u$, where $v_u$ denotes the PRM score of the $u$-th step transition. Completed trajectories in the final beam are then ranked by $R(\tau)$. Pruning occurs only when all candidate prefixes reach the same step boundary, ensuring that partial generations are not interrupted mid-step.

For *Lookahead Search*, intra-iteration control operates at a partial prefix $\tau_{1:t}$. Candidate next-step actions are evaluated by rolling out short, depth-$d$ continuations using step-delimited generation. Each rollout is segmented into intermediate steps, each step transition is scored by the PRM, and the rollout score is computed as the mean over these step-level scores. The highest-scoring continuation is selected, after which only the chosen next step is *committed*, and generation proceeds from the updated prefix. Unselected rollouts are discarded at the step boundary, and generation continues from the committed prefix without revisiting discarded branches. This process repeats at subsequent step boundaries until the iteration reaches the maximum depth.

*2) Inter-iteration selection.* Across $K$ independent iterations, each iteration produces one completed trajectory $\hat{\tau}_i$. The final output is chosen by maximizing aggregated step validity across iterations:

$$\hat{\tau}_i = \arg\max_{\tau \in \mathcal{S}_i} R(\tau), \ \hat{i} = \arg\max_{i \in \{1,\dots,K\}} R(\hat{\tau}_i), \ \hat{y} = y(\hat{\tau}_{\hat{i}}).$$
(2)

Here, $\mathcal{S}_i$ denotes the set of candidate trajectories generated within iteration $i$, and $\hat{i}$ indexes the selected iteration. In the qualitative examples as shown in Appendix A.7, the *Selected Iteration Index* denotes which iteration is chosen during inter-iteration selection (e.g., iteration 1 of $K=10$). The displayed reasoning steps correspond to the step-delimited trajectory generated within that selected iteration via intra-iteration control (e.g., Lookahead Search).

**Compute Cost Metrics for Efficiency Analysis.** To evaluate accuracy-efficiency trade-offs independently of hardware and wall-clock time, we report two complementary inference-time compute metrics that explicitly account for *all* model executions performed during adaptive inference, including controller decisions and verification (see Tables 3–4). *Theoretical FLOPs* are defined as

$$F_{\text{theo}} = \sum_{j \in \mathcal{M}} 2 M_j \cdot \min\big(\bar{T}_{\text{total}}^{(j)}, L_{\text{ctx}}^{(j)}\big) \cdot \bar{G}^{(j)},$$

where the sum runs over all models invoked at inference time, namely the base reasoning LLM, the LLM-based controller modules ($\mathcal{A}_P, \mathcal{A}_T, \mathcal{A}_C$), and the PRM. Here, $M_j$ denotes the parameter count of model $j$, $\bar{G}^{(j)}$ the average

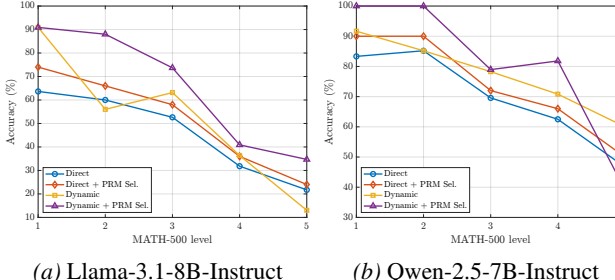

| *(a)* Llama-3.1-8B-Instruct | *(b)* Qwen-2.5-7B-Instruct |

*Figure 2.* Per-level accuracy on MATH-500 across difficulty levels.

number of forward passes of model $j$ per problem, and $\bar{T}_{\text{total}}^{(j)}$ the average total tokens processed per forward pass, capped by the model context length $L_{\text{ctx}}^{(j)}$. For example, $\bar{G}^{(\text{base})} = N$ for Best-of-$N$ sampling, while $\bar{G}^{(\text{PRM})}$ counts all prefix- and trajectory-level PRM evaluations across iterations. This formulation ensures that raw compute from controller decisions and verification is fully included. While $F_{\text{theo}}$ captures overall computational scale, it does not distinguish effective reasoning from redundant work. To quantify compute *utilization*, we define the *compute intensity score*

$$S_{\text{CI}} = \frac{\bar{G}_{\text{base}} \, \bar{T}_{\text{base}} \, (1 + \alpha\bar{C})}{\kappa},$$

where $\bar{G}_{\text{base}}$ and $\bar{T}_{\text{base}}$ denote the number of base-model generations and generated tokens per generation, respectively, and $\bar{C}$ is the average number of auxiliary model invocations, i.e., controller calls and PRM evaluations per problem. The factor $(1 + \alpha\bar{C})$ explicitly penalizes verification and control overhead, with $\alpha = 0.1$ reflecting the smaller per-call cost of these auxiliary models relative to full reasoning generations, and $\kappa = 10^6$ a normalization constant. Lower $S_{\text{CI}}$ therefore indicates that accuracy is achieved with fewer redundant reasoning tokens and limited auxiliary overhead, reflecting better concentration of computation on high-utility trajectories.

Together, $F_{\text{theo}}$ and $S_{\text{CI}}$ provide complementary views: $F_{\text{theo}}$ measures the total inference-time compute across all models involved, while $S_{\text{CI}}$ measures how efficiently base-model reasoning tokens are utilized after accounting for controller and verification overhead. This explicit accounting enables fair and transparent efficiency comparisons between adaptive and fixed inference strategies.

## 3. Experimental Setup

This section presents datasets, models, and an evaluation protocol, followed by experimental configurations and baselines.

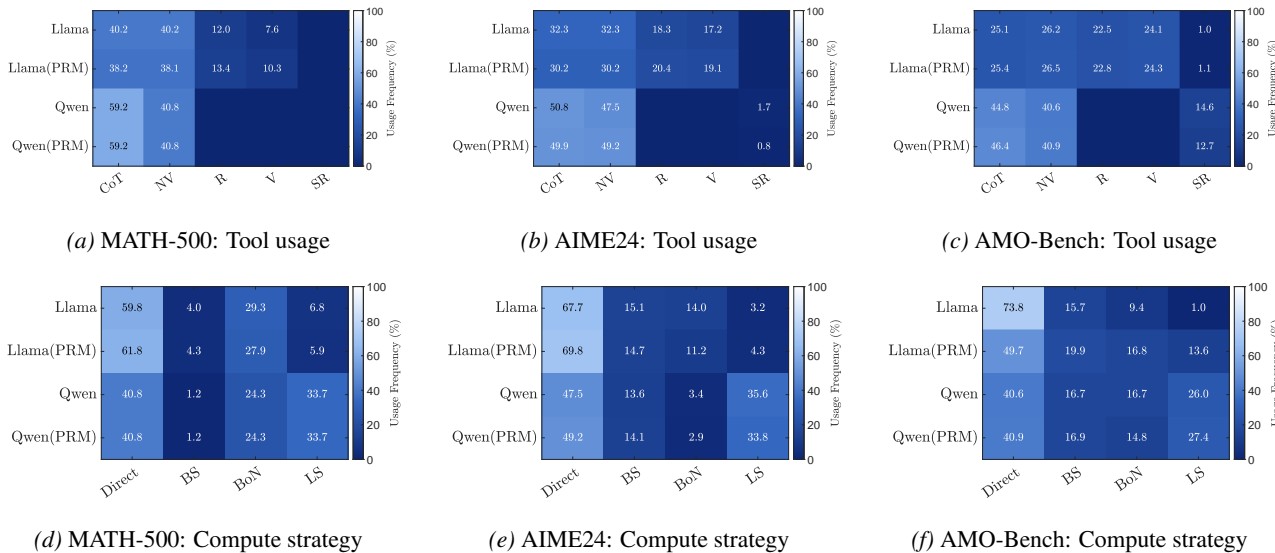

*(a) MATH-500: Tool usage*  *(b) AIME24: Tool usage*  *(c) AMO-Bench: Tool usage*

*(d) MATH-500: Compute strategy*  *(e) AIME24: Compute strategy*  *(f) AMO-Bench: Compute strategy*

*Figure 3.* **Dynamic agent configuration statistics.** Usage frequency (%) of reasoning tools and compute strategies selected by the adaptive controller across datasets. Dataset-specific patterns reflect differences in problem structure and reasoning difficulty.

### 3.1. Datasets, Models, and Evaluation Protocol

We evaluate on three mathematical reasoning benchmarks: **MATH-500** (Hendrycks et al., 2021), **AIME24**, and **AMO-Bench** (An et al., 2025). We run experiments with **Llama-3.1-8B-Instruct** (Grattafiori et al., 2024) and **Qwen-2.5-7B-Instruct** (Team, 2024). For each problem, we execute $K=10$ independent agent iterations implemented using the same base LLM, with role-specific templates to select tools ($\mathcal{A}_T$) and compute strategies ($\mathcal{A}_C$). All reasoning trajectories are scored by the pre-trained process reward model *Qwen2.5-Math-PRM-7B* (Zhang et al., 2025), and the final prediction is selected by maximizing the mean PRM reward $R_{\text{mean}}$ across iterations.

The base reasoning model uses stochastic decoding with temperature $T = 0.7$, top-$p = 0.9$, and a maximum generation length of 1024 tokens. PRM uses deterministic greedy decoding with temperature $T = 0$. All decoding settings are fixed across experiments. We adopt the math evaluation code and protocol of Satori (Shen et al., 2025). The main tables report single-run accuracies. Reporting mean accuracy and standard deviation over multiple seeds would further strengthen the robustness analysis.

### 3.2. Experimental Configurations and Baselines

We evaluate four configurations that progressively introduce adaptivity.

**(i)** *Direct* performs single-pass generation and serves as a lower-bound baseline.

**(ii)** *Direct + PRM Selection* runs $K=10$ independent di-rect generations and applies PRM-based iteration selection, isolating the effect of multi-iteration evaluation without adaptive configuration. Concretely, Direct + PRM applies the PRM solely for inter-trajectory re-ranking via Best-of-$N$ selection across $K$ completed trajectories using $R_{\text{mean}}(\tau)$ (as in Eq. 1), with no intra-trajectory PRM intervention dur-ing generation, corresponding to the best fixed strategy in Table 2.

**(iii)** *Dynamic* uses an LLM controller to select tools and compute strategies within a single iteration, isolating the benefit of adaptive configuration without multi-iteration exploration.

**(iv)** *Dynamic + PRM Selection* combines adaptive tool and compute selection with $K=10$ iterations and PRM-based selection. Overall accuracy is reported in Table 1, inference-time compute metrics $F_{\text{theo}}$ and $S_{\text{CI}}$ are shown in Figure 4, per-difficulty accuracy across Levels 1–5 on the MATH-500 dataset is reported in Figure 2, and configuration statistics summarizing tool and compute strategy usage are shown in Figure 3. In addition, we conduct fixed-configuration ablations that apply specific tool–compute–parameter com-binations uniformly across all problems and iteration counts, enabling controlled comparisons that isolate the contribution of adaptive selection, as shown in Tables 2 and 9. Further ablation results and detailed efficiency analyses are reported in Appendix A.3 and Appendix A.6.

All experiments were run on a single NVIDIA RTX 5090 GPU with 32 GB of memory. We use the same hardware for all settings and report hardware-agnostic compute metrics, i.e., theoretical FLOPs and compute intensity, rather than wall-clock time, to focus on compute utilization.

# 4. Results

Our results support three takeaways. **(i)** First, **adaptive inference**, combining dynamic tool and compute selection with PRM-guided trajectory selection, consistently outperforms uniform test-time scaling *across all evaluated datasets*. **(ii)** Second, these gains are **difficulty-dependent**: on the **MATH-500** benchmark, which is organized into five progressively harder difficulty levels, adaptive methods yield strong improvements on Levels 1–4, while gains on Level 5 are weaker due to less reliable verification signals at the highest difficulty. **(iii)** Third, adaptivity improves **compute utilization** by selectively allocating inference-time computation to high-utility reasoning trajectories rather than uniformly expanding search.

## 4.1. Adaptive Configuration Over Fixed Strategies

Table 1 shows that adaptive configuration consistently outperforms fixed inference strategies across benchmarks. PRM-based selection alone yields only limited gains when the reasoning strategy is fixed, whereas dynamically selecting tools and compute strategies leads to substantially larger improvements by tailoring reasoning to problem structure. The full Dynamic + PRM Selection setting achieves the strongest results, with large gains on MATH-500, consistent improvements on AIME24, and smaller but stable gains on AMO-Bench. Figure 2 shows that these improvements on MATH-500 are concentrated on Levels 1–4, where verification reliably filters incorrect reasoning, while on Level 5, PRM-guided multi-iteration selection partially recovers performance. Overall, these results indicate that adaptive inference is most effective when it actively configures how reasoning is performed and uses PRM-guided iteration to correct early strategic errors on harder problems.

*Table 1.* Main accuracy results on MATH-500, AIME24, and AMO-Bench.

| Dataset | Setting | Accuracy (%) | |
| --- | --- | --- | --- |
| | | Llama-3.1-8B | Qwen-2.5-7B |
| **MATH-500** | Direct | 43.8 | 71.2 |
| | + PRM Sel. | 44.6 | 72.0 |
| | Dynamic | 47.2 | 74.2 |
| | + PRM Sel. | **65.4** | **81.4** |
| **AIME24** | Direct | 3.3 | 6.67 |
| | + PRM Sel. | 6.67 | 6.67 |
| | Dynamic | 6.67 | 10.0 |
| | + PRM Sel. | **10.0** | **13.3** |
| **AMO-Bench** | Direct | 2.0 | 2.0 |
| | + PRM Sel. | 2.0 | 2.0 |
| | Dynamic | **4.0** | **4.0** |
| | + PRM Sel. | **4.0** | **4.0** |

## 4.2. Tool and Compute Strategy Selection Patterns

Figure 3 illustrates how the agent adapts reasoning tools and computation strategies under dynamic configuration. Tool usage shows clear model-dependent behavior: Llama spreads decisions across multiple tools, with CoT and Numeric Verifier dominating on MATH-500 ($\sim$40% each) but usage becoming more evenly distributed across CoT, NV, Reframer, and Verifier on harder benchmarks such as AIME24 and AMO-Bench ($\sim$17–32% each), whereas Qwen concentrates almost entirely on structured reasoning plus numeric checking (CoT $\sim$45–59% and Numeric Verifier $\sim$41–48%), with negligible use of Reframer, Verifier, or Self-Reflection except on AMO-Bench where SR reaches $\sim$15%. These patterns remain similar from 1 to 10 iterations, suggesting that *tool selection is driven by model-specific reasoning* characteristics rather than repeated sampling. Compute strategy usage is likewise complementary: Llama favors direct generation for a majority of instances ($\sim$60%), while Qwen allocates more mass to exploration strategies such as Best-of-$N$ and Lookahead (each often in the $\sim$15–35% range). Overall, these results indicate that dynamic configuration adapts inference behavior to model structure, steering each model toward reasoning paths that better align with its strengths rather than enforcing a single fixed inference recipe. A more detailed per-problem-type breakdown is provided in Appendix A.4, showing that the selected tools and compute strategies remain stable across Dynamic and Dynamic + PRM variants and vary mainly with model and problem structure.

## 4.3. Comprehensive Ablation: Fixed Configurations

Table 2 reports comprehensive ablations on MATH-500 with Qwen-2.5-7B-Instruct across all combinations of tools, compute strategies, exploration parameters ($p \in \{1, 5, 10\}$), and iteration counts ($I \in \{1, 5, 10\}$), yielding 36 fixed configurations. This grid isolates the contribution of each component and provides controlled baselines for evaluating our adaptive framework.

**Key findings.** **(i)** *Iteration count matters*: Increasing the number of iterations yields substantial accuracy gains for most fixed configurations. Performance improves markedly from a single iteration to a moderate number of iterations, moving from roughly low 70% accuracy to around 80%, and then shows diminishing returns. This trend is illustrated in Table 2 and suggests that repeated attempts help correct early reasoning errors, but iteration alone is insufficient beyond a certain point. **(ii)** *Tool choice matters*: At low iteration counts, Self-Reflection tends to outperform Chain-of-Thought, indicating that explicit self-critique is particularly valuable when only a few attempts are available. As the iteration count increases, the performance gap narrows, and both tools converge to similar accuracy levels around 80%,

*Table 2.* Ablation study: Accuracy (%) on MATH-500 with Qwen-2.5-7B-Instruct across fixed tool–strategy–parameter–iteration configurations.

| Param | Iter | CoT | | | Self-Reflection | | |
|---|---|---|---|---|---|---|---|
| | | **Best-of-N** | **Beam Search** | **Lookahead** | **Best-of-N** | **Beam Search** | **Lookahead** |
| | 1 | 71.2 | 70.4 | 70.2 | 75.6 | 75.2 | 71.8 |
| 1 | 5 | **81.4** | **81.4** | **81.4** | 80.2 | 80.6 | 79.4 |
| | 10 | 81.2 | 81.2 | 81.2 | 80.8 | 80.4 | 79.2 |
| | 1 | 75.8 | 76.2 | 75.2 | 74.2 | 76.8 | 75.6 |
| 5 | 5 | **81.4** | 79.4 | 80.8 | 79.4 | 79.2 | 79.8 |
| | 10 | **81.4** | 79.8 | 80.6 | 79.8 | 79.6 | 79.4 |
| | 1 | 72.2 | 73.6 | 72.4 | 72.6 | 74.4 | 74.2 |
| 10 | 5 | **81.4** | 80.2 | 80.2 | 79.2 | 79.8 | 79.6 |
| | 10 | **81.4** | 80.6 | 80.8 | 79.6 | 79.4 | 79.8 |

as selection mitigates individual trajectory failures. These patterns are shown in Table 2. **(iii)** *Strategy–parameter interactions are non-trivial*: Exploration parameters exhibit non-monotonic effects. Moderate exploration improves accuracy, whereas more aggressive exploration can reduce performance at low iteration counts, indicating that excessive branching introduces noise when not paired with sufficient iteration or selection. This behavior is documented in Table 2. **(iv)** *No single fixed configuration dominates*: Carefully tuned fixed configurations can reach strong performance, up to approximately 81% accuracy on MATH-500, but doing so requires prior knowledge of the optimal tool, strategy, and parameter combination. In contrast, the adaptive method reaches comparable performance automatically by selecting configurations on a per-problem basis, as shown in Table 2 and further analyzed in Appendix A.2.

**Compute efficiency.** Dynamic + PRM Selection attains 81.4% accuracy while concentrating computation on high-utility trajectories and terminating weak ones early, resulting in substantially lower compute intensity despite slightly higher theoretical FLOPs. In contrast, as detailed in Appendix A.2, achieving the same accuracy with fixed configurations requires $1.31 \times 10^{14}$ FLOPs and a compute intensity of $7.97 \times 10^{-3}$ using CoT with Best-of-$N$ sampling at $p=10$ and $I=10$, reflecting significant wasted computation on easier instances.

Table 9 reports corresponding ablations on AIME24. Unlike MATH-500, no fixed configuration exceeds 10.0% accuracy (best: CoT Lookahead, $p=5$, $I \in \{1, 5, 10\}$), compared to 13.3% for Dynamic + PRM Selection. Increasing $p$ degrades performance, for example, from 6.67% to 10.0% and then to 3.3% for CoT Lookahead as $p$ increases, and gains from increasing $I$ are marginal. This indicates that trajectory diversity from adaptive multi-tool selection, rather than uniform repetition, is essential for highly challenging problems.

### 4.4. Compute Cost Scaling and Accuracy–Efficiency Trade-offs

Figure 4 illustrates how inference-time compute scales across increasingly adaptive configurations, measured by theoretical FLOPs and $S_{CI}$. As adaptivity increases from Direct inference to Dynamic + PRM Selection, both FLOPs and $S_{CI}$ grow monotonically due to additional controller calls, verification, and multi-iteration exploration. This figure therefore reflects *raw compute scaling* rather than efficiency at matched accuracy: configurations are compared at fixed inference settings, and while Direct inference incurs lower computational cost, it also achieves substantially lower accuracy (e.g., 43.8% vs. 65.4% on MATH-500 for Llama-3.1-8B-Instruct).

Our efficiency claim is accuracy-conditional: rather than minimizing $S_{CI}$ in absolute terms, Dynamic + PRM Selection achieves higher accuracy for a given compute budget and matches the accuracy of strong fixed strategies with better utilization. As shown in Figure 5, fixed configurations reach ~81% accuracy only by sharply increasing $S_{CI}$ through uniform search scaling, whereas Dynamic + PRM Selection attains the same accuracy with lower $S_{CI}$. This indicates that adaptive inference reallocates additional compute toward high-utility reasoning trajectories rather than redundant generation, improving compute utilization conditional on accuracy, which is the central objective of adaptive test-time compute allocation.

## 5. Limitations and Future Directions

A primary limitation is reliance on PRM quality. While PRM-guided selection improves performance overall, especially on easier problems, its effectiveness decreases on the hardest problems, where the PRM may misrank plausible but incorrect trajectories, as shown in Table 1 and Figure 2. PRMs can be over-confident or prefer locally correct but globally wrong reasoning paths; we mitigate this

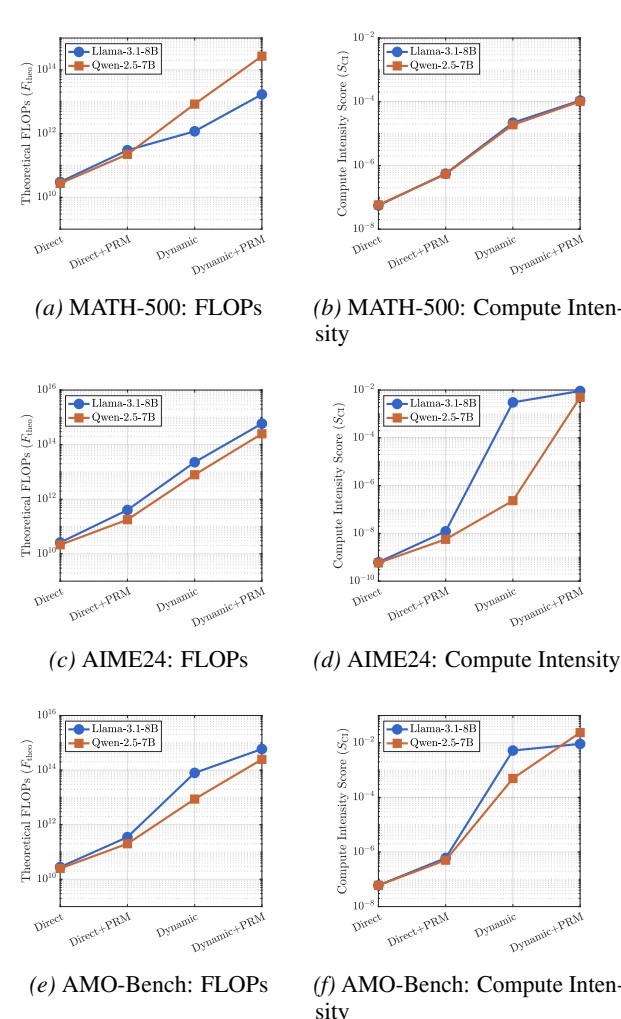

*(a) MATH-500: FLOPs*   *(b) MATH-500: Compute Intensity*

*(c) AIME24: FLOPs*   *(d) AIME24: Compute Intensity*

*(e) AMO-Bench: FLOPs*   *(f) AMO-Bench: Compute Intensity*

*Figure 4.* **Compute cost scaling across datasets.** Theoretical FLOPs and compute intensity ($S_{CI}$) for main experimental configurations on MATH-500, AIME24, and AMO-Bench.

with ternary scoring and diverse $K=10$ iterations across tools and strategies. These limitations motivate difficulty-aware verification, targeted supervision on hard instances, curriculum-based PRM training, and future multi-PRM ablations, such as using Math-Shepherd (Wang et al., 2024), to assess robustness across verifier choices. The heuristic controller enables training-free adaptivity, but future work should explore learned, budget-aware policies for tool use, compute allocation, and stopping decisions (Paglieri et al., 2025; Liu et al., 2025). Because the controller uses role-conditioned prompts, its reliability may depend on the base model's instruction-following ability, especially for smaller models. Multi-branch search also increases latency, but batching and parallel execution can reduce this overhead, as discussed in Appendix A.5. Future work should study prompt sensitivity, smaller models, and latency-aware deployment. Finally, generalization beyond mathematical

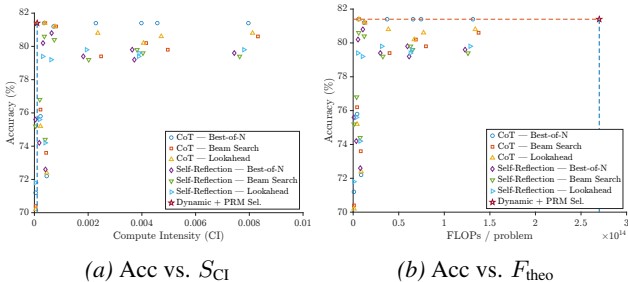

*(a) Acc vs. $S_{CI}$*   *(b) Acc vs. $F_{theo}$*

*Figure 5.* Accuracy-efficiency trade-offs on MATH-500 (Qwen-2.5-7B).

reasoning remains open and will require domain-adaptive verification signals and broader evaluation.

## 6. Related Work

**Test-Time Compute Scaling.** Scaling inference-time computation improves reasoning performance more effectively than increasing model size (Snell et al., 2025). Test-time compute scaling methods, including repeated sampling, beam search, chain-of-thought prompting, and verification-based selection, have substantially improved performance on reasoning-intensive tasks, particularly in mathematics (Wei et al., 2022; Lightman et al., 2024; Brown et al., 2024). Recent agentic approaches further integrate verification with structured search, such as Monte Carlo tree search with preference learning and self-refinement (Brown et al., 2024; Rakhsha et al., 2025; Xie et al., 2024; Zhang et al., 2024; Guan et al., 2025), but typically rely on fixed inference procedures (e.g., predetermined Best-of-$N$ or search depth), limiting flexibility across strategies. Complementary work explores adaptive allocation via difficulty estimation under fixed budgets (Du, 2025) or budget-aware tool invocation (Liu et al., 2025; Paglieri et al., 2025). Reward-guided approaches use collaborative generation with outcome-based signals to steer compute toward high-reward paths (Muñoz & Yuan, 2025); unlike our work, strategy and tool configuration remain fixed. More recent work studies adaptive inference policies that dynamically adjust reasoning depth, search breadth, or stopping criteria based on uncertainty or intermediate signals, rather than relying on fixed budgets or difficulty predictors (Banfi & Gamage, 2026).

**Reasoning Strategies and Verification.** Chain-of-thought prompting elicits step-by-step reasoning (Wei et al., 2022), but its benefits are domain-dependent (Sprague et al., 2025) and explanations may be unfaithful (Lanham et al., 2023; Turpin et al., 2023). PRMs evaluate intermediate reasoning steps (Lightman et al., 2024), while LLM-based critics and interactive self-improvement methods support error detection and correction (McAleese et al., 2024; Yu et al., 2024). Trajectory-level extensions such as ReasonFlux-

PRM scale verification to long reasoning traces (Zou et al., 2025), and symbolic orchestration approaches (e.g., Core-Think) introduce explicit control layers with structured verification (Vaghasiya et al., 2025). Subsequent work extends PRMs to reflective and generative settings (Zhao et al., 2025), studying their behavior on self-correction traces (Yang et al., 2025), tool-augmented reasoning, and long-horizon verification (Agarwal et al., 2025), while also highlighting limitations of PRM reliability on complex problems.

**Reasoning as Decision-Making.** Recent work frames LLM reasoning as sequential decision-making, using reinforcement learning for trial-and-error learning (Havrilla et al., 2024) and counterfactual analyses to characterize genuine reasoning behavior (Wu et al., 2024).

# 7. Conclusion

This work shows that verification-guided adaptive test-time inference more effectively converts compute into accuracy by dynamically shaping the reasoning process. Guided by a PRM during generation, the approach improves reasoning CoT steps beyond fixed strategies, particularly on easier to moderately difficult problems. Overall, the results show that adaptivity outperforms uniform repetition and that online verification is more effective than post hoc reranking, making adaptive, verifier-guided inference a practical and model-agnostic path to compute-efficient and reliable reasoning.

While our study focuses on mathematical reasoning, the core idea is domain-agnostic: using intermediate verification signals to dynamically choose tools, allocate compute, and stop early when appropriate. This framework can extend to domains such as code generation, long-context QA, and safety-critical decision-making when tasks provide clear intermediate steps, reliable non-terminal verification signals, and heterogeneous difficulty across instances. Examples include unit-test feedback for code and entailment checks for multi-hop QA. In domains with only sparse final-answer rewards, effective deployment may require domain-adaptive verifiers or PRM training under latency and compute constraints.

# Impact Statement

By allocating test-time compute adaptively rather than uniformly, our framework provides a foundation for more efficient and reliable LLM reasoning. Its training-free design and modular verification interface make it broadly applicable across domains where compute efficiency and reasoning quality are critical.

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

# A. Supplementary Materials

## A.1. Compute Cost Metrics

*Table 3.* Compute cost metrics for evaluating test-time efficiency.

| Metric | Symbol | Definition |
|--------|--------|------------|
| Theoretical FLOPs | $F_{\text{theo}}$ | $\displaystyle\sum_{j\in\mathcal{M}} 2\,M_j\,T_{\text{forward}}^{(j)}\,\bar{G}^{(j)}$ |
| Compute Intensity Score | $S_{\text{CI}}$ | $\dfrac{\bar{G}_{\text{base}}\,\bar{T}_{\text{base}}\,(1+\alpha\bar{C})}{\kappa}$ |

*Table 4.* Notation used in compute cost metrics.

| Symbol | Description |
|--------|-------------|
| $\mathcal{M}$ | Set of models invoked at inference time (base LLM, controller LLM, PRM) |
| $M_j$ | Number of parameters of model $j \in \mathcal{M}$ |
| $\bar{G}^{(j)}$ | Average number of forward passes of model $j$ per problem |
| $\bar{T}_{\text{total}}^{(j)}$ | Average total tokens processed per forward pass of model $j$ |
| $T_{\text{forward}}^{(j)}$ | $\min(\bar{T}_{\text{total}}^{(j)}, L_{\text{ctx}}^{(j)})$ |
| $L_{\text{ctx}}^{(j)}$ | Maximum context length of model $j$ |
| $\bar{G}_{\text{base}}$ | Average number of base-model generations per problem |
| $\bar{T}_{\text{base}}$ | Average generated tokens per base-model generation |
| $\bar{C}$ | Average number of auxiliary model calls (controller + PRM) per problem |
| $\alpha$ | Fixed overhead factor per auxiliary call (set to 0.1) |
| $\kappa$ | Normalization constant for $S_{\text{CI}}$ (e.g., $10^6$) |

## A.2. Compute Efficiency vs. Raw Compute Trade-off

Dynamic + PRM can exhibit slightly lower compute intensity ($S_{\text{CI}}$) while incurring modestly higher FLOPs because $S_{\text{CI}}$ measures *how efficiently computation contributes to correct reasoning*, rather than raw computational amount alone. In fixed baselines, a predetermined number of iterations or beams are executed for every problem, leading to substantial wasted computation on easy instances or low-quality reasoning trajectories, which increases $S_{\text{CI}}$.

Tables 5 and 6 illustrate this phenomenon across all 36 fixed configurations. For example, CoT Beam Search with $(p = 5, I = 5)$ achieves 79.4% accuracy but requires $3.99 \times 10^{13}$ FLOPs per problem with high $S_{CI} = 2.48 \times 10^{-3}$, while CoT Best-of-N with $(p = 10, I = 5)$ reaches 81.4% accuracy at even larger cost: $6.57 \times 10^{13}$ FLOPs and $S_{CI} = 3.98 \times 10^{-3}$. This trend indicates that a substantial fraction of computation in fixed settings is spent on low-utility trajectories, especially for easier instances, inflating $S_{\text{CI}}$ without proportional accuracy gains.

In contrast, the dynamic framework leverages PRM scoring to adaptively select and prioritize high-utility trajectories and terminate or deprioritize weak ones early, so a larger fraction of the compute budget directly supports correct reasoning, resulting in lower $S_{\text{CI}}$. This adaptivity introduces modest overhead from candidate generation, PRM evaluation, and selection or verification steps, slightly increasing total FLOPs. However, this tradeoff is favorable: Dynamic + PRM trades a small increase in raw compute for substantially improved compute utilization, enabling comparable or higher accuracy with better efficiency. This effect is consistent with Figure 5, which shows that higher accuracy in fixed baselines is achieved by moving to regions of sharply increased FLOPs and $S_{\text{CI}}$, whereas PRM-guided allocation shifts computation toward more efficient regions of the accuracy-FLOPs plane.

## A.3. Detailed Ablation Results: Compute Costs

Tables 5 and 6 provide detailed compute cost breakdowns for all 36 fixed configurations on MATH-500 with Qwen-2.5-7B-Instruct, corresponding to the accuracy results in Table 2 (main text). These tables enable analysis of accuracy-efficiency trade-offs across the full configuration space.

**Key observations**: (1) Both $F_{\text{theo}}$ and $S_{\text{CI}}$ scale roughly linearly with iteration count for fixed parameter settings. (2) Higher

*Table 5.* Compute Intensity ($S_{\text{CI}}$) for all fixed configurations on MATH-500 (Qwen-2.5-7B-Instruct).

| Param | Iter | CoT | | | Self-Reflection | | |
|---|---|---|---|---|---|---|---|
| | | **Best-of-N** | **Beam Search** | **Lookahead** | **Best-of-N** | **Beam Search** | **Lookahead** |
| 1 | 1 | $4.35 \times 10^{-5}$ | $4.57 \times 10^{-5}$ | $4.52 \times 10^{-5}$ | $3.99 \times 10^{-5}$ | $4.65 \times 10^{-5}$ | $3.91 \times 10^{-5}$ |
| | 5 | $3.57 \times 10^{-4}$ | $4.04 \times 10^{-4}$ | $3.80 \times 10^{-4}$ | $3.20 \times 10^{-4}$ | $3.70 \times 10^{-4}$ | $3.10 \times 10^{-4}$ |
| | 10 | $7.14 \times 10^{-4}$ | $8.08 \times 10^{-4}$ | $7.60 \times 10^{-4}$ | $6.40 \times 10^{-4}$ | $7.40 \times 10^{-4}$ | $6.20 \times 10^{-4}$ |
| 5 | 1 | $2.26 \times 10^{-4}$ | $2.26 \times 10^{-4}$ | $2.25 \times 10^{-4}$ | $1.81 \times 10^{-4}$ | $2.00 \times 10^{-4}$ | $1.90 \times 10^{-4}$ |
| | 5 | $2.29 \times 10^{-3}$ | $2.48 \times 10^{-3}$ | $2.36 \times 10^{-3}$ | $1.82 \times 10^{-3}$ | $2.02 \times 10^{-3}$ | $1.94 \times 10^{-3}$ |
| | 10 | $4.57 \times 10^{-3}$ | $4.96 \times 10^{-3}$ | $4.72 \times 10^{-3}$ | $3.64 \times 10^{-3}$ | $4.04 \times 10^{-3}$ | $3.88 \times 10^{-3}$ |
| 10 | 1 | $4.56 \times 10^{-4}$ | $4.37 \times 10^{-4}$ | $4.58 \times 10^{-4}$ | $4.10 \times 10^{-4}$ | $4.00 \times 10^{-4}$ | $4.00 \times 10^{-4}$ |
| | 5 | $3.98 \times 10^{-3}$ | $4.16 \times 10^{-3}$ | $4.06 \times 10^{-3}$ | $3.72 \times 10^{-3}$ | $3.82 \times 10^{-3}$ | $3.90 \times 10^{-3}$ |
| | 10 | $7.97 \times 10^{-3}$ | $8.32 \times 10^{-3}$ | $8.12 \times 10^{-3}$ | $7.44 \times 10^{-3}$ | $7.64 \times 10^{-3}$ | $7.80 \times 10^{-3}$ |

*Table 6.* Theoretical FLOPs ($F_{\text{theo}}$) per problem for all fixed configurations on MATH-500 (Qwen-2.5-7B-Instruct).

| Param | Iter | CoT | | | Self-Reflection | | |
|---|---|---|---|---|---|---|---|
| | | **Best-of-N** | **Beam Search** | **Lookahead** | **Best-of-N** | **Beam Search** | **Lookahead** |
| 1 | 1 | $8.30 \times 10^{11}$ | $8.72 \times 10^{11}$ | $8.63 \times 10^{11}$ | $7.61 \times 10^{11}$ | $8.88 \times 10^{11}$ | $7.46 \times 10^{11}$ |
| | 5 | $5.93 \times 10^{12}$ | $6.64 \times 10^{12}$ | $6.30 \times 10^{12}$ | $5.32 \times 10^{12}$ | $6.08 \times 10^{12}$ | $5.12 \times 10^{12}$ |
| | 10 | $1.19 \times 10^{13}$ | $1.33 \times 10^{13}$ | $1.26 \times 10^{13}$ | $1.06 \times 10^{13}$ | $1.22 \times 10^{13}$ | $1.02 \times 10^{13}$ |
| 5 | 1 | $4.32 \times 10^{12}$ | $4.32 \times 10^{12}$ | $4.29 \times 10^{12}$ | $3.45 \times 10^{12}$ | $3.81 \times 10^{12}$ | $3.62 \times 10^{12}$ |
| | 5 | $3.73 \times 10^{13}$ | $3.99 \times 10^{13}$ | $3.86 \times 10^{13}$ | $2.98 \times 10^{13}$ | $3.26 \times 10^{13}$ | $3.14 \times 10^{13}$ |
| | 10 | $7.45 \times 10^{13}$ | $7.99 \times 10^{13}$ | $7.72 \times 10^{13}$ | $5.96 \times 10^{13}$ | $6.52 \times 10^{13}$ | $6.28 \times 10^{13}$ |
| 10 | 1 | $8.71 \times 10^{12}$ | $8.35 \times 10^{12}$ | $8.74 \times 10^{12}$ | $7.83 \times 10^{12}$ | $7.63 \times 10^{12}$ | $7.64 \times 10^{12}$ |
| | 5 | $6.57 \times 10^{13}$ | $6.88 \times 10^{13}$ | $6.70 \times 10^{13}$ | $6.14 \times 10^{13}$ | $6.30 \times 10^{13}$ | $6.42 \times 10^{13}$ |
| | 10 | $1.31 \times 10^{14}$ | $1.38 \times 10^{14}$ | $1.34 \times 10^{14}$ | $1.23 \times 10^{14}$ | $1.26 \times 10^{14}$ | $1.28 \times 10^{14}$ |

exploration parameters ($p$) increase computational costs at single iterations but show diminishing efficiency returns at higher iterations. (3) Self-Reflection configurations generally exhibit lower compute costs than CoT at comparable accuracy levels, suggesting more efficient reasoning for this model. (4) The most computationally expensive configuration (CoT Beam Search, $p = 10$, $I = 10$) requires $1.38 \times 10^{14}$ FLOPs with $S_{CI} = 8.32 \times 10^{-3}$, substantially higher than Dynamic + PRM while achieving identical 81.4% accuracy, validating our framework's efficiency advantage.

### A.4. Per-Problem Strategy Distribution

Table 7 reports tool and strategy choices by MATH-500 problem type. Llama uses CoT and Numeric Verifier broadly, with more Reframer use on harder types, while Qwen mainly uses CoT, Numeric Verifier, and Lookahead.

### A.5. Latency

We report hardware-agnostic metrics ($F_{\text{theo}}$, $S_{\text{CI}}$) in the main text, while Table 8 summarizes hardware-specific latency on a single NVIDIA RTX 5090 GPU. Sequential latency grows roughly with $K$, whereas parallel execution reduces the effective cost to about $1.5$–$2\times$ direct generation. Sequential latency scales roughly linearly with $K$, while parallel execution across iterations reduces the effective cost to about $1.5$–$2\times$ a single direct generation.

*Table 7.* Tool and compute-strategy distributions by MATH-500 problem type. Values are percentages.

| Problem Type | CoT | NV | R | V | SR | Direct | BS | BoN | LS |
|---|---|---|---|---|---|---|---|---|---|
| **Llama-3.1-8B** | | | | | | | | | |
| Algebra | 41.5 | 41.5 | 9.5 | 7.5 | 0 | 61.0 | 4.0 | 29.0 | 6.0 |
| Counting & Probability | 39.5 | 39.5 | 13.5 | 7.5 | 0 | 59.0 | 4.0 | 30.0 | 7.0 |
| Geometry | 40.0 | 40.0 | 12.5 | 7.5 | 0 | 60.0 | 4.0 | 29.5 | 6.5 |
| Intermediate Algebra | 38.5 | 38.5 | 15.5 | 7.5 | 0 | 57.5 | 4.5 | 31.5 | 6.5 |
| Number Theory | 39.5 | 39.5 | 13.5 | 7.5 | 0 | 58.5 | 4.5 | 30.5 | 6.5 |
| Prealgebra | 42.0 | 42.0 | 9.5 | 6.5 | 0 | 63.0 | 4.0 | 27.0 | 6.0 |
| Precalculus | 40.0 | 40.0 | 11.5 | 8.5 | 0 | 61.0 | 3.5 | 29.0 | 6.5 |
| *Mean* | *40.1* | *40.1* | *12.2* | *7.5* | *0* | *60.0* | *4.1* | *29.5* | *6.4* |
| **Llama-3.1-8B + PRM** | | | | | | | | | |
| Algebra | 39.5 | 39.5 | 12.5 | 8.5 | 0 | 63.5 | 4.0 | 27.0 | 5.5 |
| Counting & Probability | 37.0 | 37.0 | 14.5 | 11.5 | 0 | 60.5 | 4.5 | 28.5 | 6.5 |
| Geometry | 37.5 | 37.5 | 14.0 | 11.0 | 0 | 61.5 | 4.5 | 28.0 | 6.0 |
| Intermediate Algebra | 36.0 | 36.0 | 17.5 | 10.5 | 0 | 59.0 | 4.5 | 30.5 | 6.0 |
| Number Theory | 36.0 | 36.0 | 16.5 | 11.5 | 0 | 60.0 | 4.5 | 29.5 | 6.0 |
| Prealgebra | 41.5 | 41.5 | 9.5 | 7.5 | 0 | 65.5 | 4.0 | 26.0 | 4.5 |
| Precalculus | 39.5 | 39.5 | 10.5 | 10.5 | 0 | 63.5 | 4.0 | 27.5 | 5.0 |
| *Mean* | *38.1* | *38.1* | *13.6* | *10.1* | *0* | *61.9* | *4.3* | *28.1* | *5.6* |
| **Qwen-2.5-7B** | | | | | | | | | |
| Algebra | 60.0 | 40.0 | 0 | 0 | 0 | 41.0 | 1.0 | 25.0 | 33.0 |
| Counting & Probability | 58.0 | 42.0 | 0 | 0 | 0 | 41.0 | 1.5 | 24.0 | 33.5 |
| Geometry | 62.0 | 38.0 | 0 | 0 | 0 | 39.0 | 1.5 | 23.0 | 36.5 |
| Intermediate Algebra | 58.0 | 42.0 | 0 | 0 | 0 | 41.0 | 1.0 | 25.0 | 33.0 |
| Number Theory | 57.0 | 43.0 | 0 | 0 | 0 | 42.0 | 1.0 | 25.0 | 32.0 |
| Prealgebra | 61.0 | 39.0 | 0 | 0 | 0 | 43.0 | 1.0 | 24.0 | 32.0 |
| Precalculus | 59.0 | 41.0 | 0 | 0 | 0 | 40.0 | 1.0 | 24.0 | 35.0 |
| *Mean* | *59.3* | *40.7* | *0* | *0* | *0* | *41.0* | *1.1* | *24.3* | *33.6* |
| **Qwen-2.5-7B + PRM** | | | | | | | | | |
| Algebra | 59.0 | 41.0 | 0 | 0 | 0 | 41.0 | 1.0 | 24.0 | 34.0 |
| Counting & Probability | 58.0 | 42.0 | 0 | 0 | 0 | 41.0 | 1.5 | 24.0 | 33.5 |
| Geometry | 61.0 | 39.0 | 0 | 0 | 0 | 39.0 | 1.5 | 23.0 | 36.5 |
| Intermediate Algebra | 58.0 | 42.0 | 0 | 0 | 0 | 41.0 | 1.0 | 25.0 | 33.0 |
| Number Theory | 57.0 | 43.0 | 0 | 0 | 0 | 42.0 | 1.0 | 25.0 | 32.0 |
| Prealgebra | 61.0 | 39.0 | 0 | 0 | 0 | 43.0 | 1.0 | 24.0 | 32.0 |
| Precalculus | 60.0 | 40.0 | 0 | 0 | 0 | 40.0 | 1.0 | 24.0 | 35.0 |
| *Mean* | *59.1* | *40.9* | *0* | *0* | *0* | *41.0* | *1.1* | *24.1* | *33.7* |

*Table 8.* Wall-clock latency scaling relative to Direct inference ($K = 1$) on a single NVIDIA RTX 5090 GPU.

| Configuration | Sequential Latency | Parallel Latency |
|---|---|---|
| Direct ($K = 1$) | $1\times$ (baseline) | $1\times$ (baseline) |
| Direct + PRM Sel. ($K = 10$) | $\approx 10\times$ | $\approx 1.5$–$2\times$ |
| Dynamic ($K = 1$) | $\approx 1.3$–$1.5\times$ | $\approx 1.3$–$1.5\times$ |
| Dynamic + PRM Sel. ($K = 10$) | $\approx 10$–$12\times$ | $\approx 1.5$–$2\times$ |

## A.6. AIME24 Ablation Analysis

Table 9 presents ablation results on AIME24 with Qwen-2.5-7B-Instruct, revealing distinct patterns compared to MATH-500 due to the higher difficulty level of competition-level problems.

**Critical insights**: (1) **No single configuration dominates**: The best fixed configuration (CoT Lookahead, $p = 5$) achieves only 10.0% accuracy, substantially below Dynamic + PRM Selection's 13.3%, demonstrating that adaptive multi-tool selection is essential for extremely challenging problems. (2) **Non-monotonic compute scaling**: Increasing exploration parameters often degrades performance (e.g., CoT Lookahead: 6.67% at $p = 1 \rightarrow 10.0\%$ at $p = 5 \rightarrow 3.3\%$ at $p = 10$), indicating excessive search introduces noise on competition-level problems where solution spaces are more constrained. (3) **Complex tool-strategy interactions**: CoT performs best with moderate Lookahead (10.0%), while Self-Reflection shows stable but lower performance (3.3-6.67%), validating the need for problem-specific adaptive tool selection. (4) **Limited iteration benefits for fixed tools**: Marginal improvements from $I = 1$ to $I = 10$ for single-tool configurations highlight that trajectory *diversity* from multiple tools, not mere repetition, drives PRM selection effectiveness, a key advantage of our dynamic framework.

*Table 9.* Ablation study: Accuracy (%) on AIME24 with Qwen-2.5-7B-Instruct across fixed configurations.

| Param | Iter | CoT | | | Self-Reflection | | |
|---|---|---|---|---|---|---|---|
| | | **Best-of-N** | **Beam Search** | **Lookahead** | **Best-of-N** | **Beam Search** | **Lookahead** |
| | 1 | 3.3 | 3.3 | 6.67 | 3.3 | 3.3 | 3.3 |
| 1 | 5 | 3.3 | 3.3 | 6.67 | 3.3 | 3.3 | 6.67 |
| | 10 | 6.67 | 6.67 | 6.67 | 3.3 | 6.67 | 6.67 |
| | 1 | 6.67 | 6.67 | 10.0 | 3.3 | 3.3 | 6.67 |
| 5 | 5 | 6.67 | 6.67 | 10.0 | 3.3 | 6.67 | 6.67 |
| | 10 | 10.0 | 6.67 | 10.0 | 6.67 | 6.67 | 6.67 |
| | 1 | 6.67 | 6.67 | 3.3 | 3.3 | 6.67 | 3.3 |
| 10 | 5 | 6.67 | 6.67 | 3.3 | 6.67 | 6.67 | 3.3 |
| | 10 | 6.67 | 6.67 | 3.3 | 6.67 | 6.67 | 6.67 |

## A.7. Illustrative Example: Direct vs. Dynamic+PRM

We illustrate the effectiveness of our framework through a representative example from the MATH-500 dataset, comparing a failed direct inference attempt (Sample A) with a successful PRM-selected iteration using dynamic tool selection and verification (Sample B).

---

**Problem Statement**

Given the system of logarithmic equations:

$$\log_x(y^x) = 10, \qquad \log_y(x^{4y}) = 10,$$

where $x, y > 0$ and $x, y \neq 1$, find the product $xy$.

---

**Direct Reasoning Inference (Sample A): Incorrect Prediction: $xy = 64$**

**Agent Configuration**

**Planning Module ($\mathcal{A}_P$):** Not used
**Tool Selection ($\mathcal{A}_T$):** Not used
**Verifiers:** None (no numeric verifier invoked)
**Compute Selection ($\mathcal{A}_C$):** Not used
**PRM Selection:** Not used

**Plan:** Determine the values of $x$ and $y$ by converting the logarithmic equations into exponential form, then solve for $xy$.
**Reasoning Trajectory:**
*Step 1: Convert to exponential form*

$$y^x = x^{10}, \qquad x^{4y} = y^{10}.$$

*Step 2: Isolate $y$ from first equation*

$$y = x^{10/x}.$$

*Step 3: Substitute into second equation*

$$x^{4y} = y^{10} \implies x^{4x^{10/x}} = (x^{10/x})^{10} = x^{100/x}.$$

*Step 4: Equate exponents (assuming $x > 1$)*

$$4x^{10/x} = \frac{100}{x} \implies x^{10/x} = \frac{25}{x} \implies x^{11/x} = 25.$$

**Critical Error:** *Unjustified assumption*: The model assumes $x = 2$ satisfies $x^{11/x} = 25$ without verification:

$$2^{11/2} = 2^{5.5} \approx 45.25 \neq 25.$$

*Step 5: Incorrect computation*
Assuming $x = 2$:

$$y = x^{10/x} = 2^{10/2} = 2^5 = 32.$$

Therefore:

$$xy = 2 \cdot 32 = 64. \quad \times \text{ INCORRECT}$$

**Final Answer:** $\boxed{64}$
**Failure Analysis:** This failure arises from single-pass direct generation without adaptive tool or compute selection. In the absence of the adaptive pipeline, the controller does not invoke verification tools (e.g., the Numeric Verifier) and no PRM-guided selection is applied. As a result, the algebraic error and unverified assumption in Step 4 remain undetected, allowing the mistake to propagate to the final answer.

> **Agentic + PRM Selection (Sample B): Correct Prediction: xy = 25**
>
> > **Agent Configuration**
> >
> > **Planning Module ($\mathcal{A}_P$):** Enabled
> > **Tool Selection ($\mathcal{A}_T$):** Chain-of-Thought (CoT) + Numeric Verifier
> > **Compute Selection ($\mathcal{A}_C$):** Lookahead Search (LA), $d = 3$
> > **Final Answer Extraction ($\mathcal{A}_F$):** Standard numeric extraction
> > **PRM Selection: YES**
>
> > **Process Reward Model (PRM) Trajectory Scoring**
> >
> > **Mean Reward ($R_{\text{mean}}$):** 0.953125
> > **Selected Iteration Index:** 1 (out of $K = 10$)
> > *This iteration was selected as the final answer from 10 candidate iterations based on highest $R_{mean}$ score.*
>
> **Plan:** Determine the values of $x$ and $y$ from the given logarithmic equations using logarithmic properties, then find the product $xy$.
>
> **Reasoning Trajectory:**
> *Step 1: Apply logarithmic power rule*
> From $\log_x(y^x) = 10$:
> $$\log_x(y^x) = x \log_x y = 10 \;\Rightarrow\; \log_x y = \frac{10}{x}.$$
>
> From $\log_y(x^{4y}) = 10$:
> $$\log_y(x^{4y}) = 4y \log_y x = 10 \;\Rightarrow\; \log_y x = \frac{10}{4y} = \frac{5}{2y}.$$
>
> *Step 2: Apply logarithmic reciprocity property*
> Since $\log_x y \cdot \log_y x = 1$ for valid $x, y$:
> $$\left(\frac{10}{x}\right)\left(\frac{5}{2y}\right) = 1.$$
>
> *Step 3: Solve for $xy$*
> $$\frac{50}{2xy} = 1 \;\Rightarrow\; \frac{25}{xy} = 1 \;\Rightarrow\; xy = 25.$$
>
> **Numeric Verification:**
>
> - **Verdict: CORRECT**
>
> - **Verification Score:** 1.0
>
> - The numeric verifier confirmed the algebraic manipulations and final answer.
>
> **Final Answer:** $\boxed{25}$ ✓ **CORRECT**
>
> **Success Analysis:** The inclusion of the numeric verifier in the tool selection and the PRM-based trajectory scoring enabled the system to: (1) identify the correct mathematical approach using logarithmic properties rather than exponential substitution, (2) verify intermediate algebraic steps, and (3) select this high-quality trajectory from multiple candidate iterations.

This example shows why the adaptive pipeline outperforms direct answer generation. By dynamically selecting verification tools and using PRM-based trajectory selection, the agent corrects errors early and prioritizes correct reasoning paths across iterations. As a result, additional compute is selectively allocated to high-utility trajectories rather than uniformly scaling computation, leading to improved solution correctness.

# B. Prompt Architecture

## B.1. System Prompts

---
**Mathematical Problem Solver System Prompt**

You are a specialized mathematical problem solver. Your role is to solve the user's math question accurately and efficiently, using appropriate intermediate reasoning (when requested by other prompts) and providing answers in the formats specified by the calling prompt.

---

## B.2. Stage 1: Planning Module ($\mathcal{A}_P$)

---
**Planning Prompt Template**

Review the user's specific math problem and create a concise high-level plan for solving it.
Output your plan in this format:
`<plan>YOUR SOLUTION APPROACH</plan>`
Be specific about the main steps you will take, but do NOT solve the problem.
Problem: `{problem}`
**\* STRICT REQUIREMENTS \***

1. DO NOT solve the problem.

2. DO NOT perform algebra, arithmetic, or simplification.

3. Write a 1–3 sentence plan only.

4. Wrap the plan EXACTLY inside `<plan>...</plan>`.

5. No text is allowed outside the `<plan>` tags.

Output format (mandatory):
`<plan>STEP-BY-STEP HIGH-LEVEL STRATEGY ONLY</plan>`

---

## B.3. Stage 2: Tool Selection ($\mathcal{A}_T$)

---
**Tool Selector System Prompt**

You are a tool selector for solving a specific math problem.
**Task:**
Given the user's math question, choose which reasoning and verification tools should be applied. You are not solving the problem yourself; you only decide which tools to run.
**Tool characteristics:**

- **self-reflection**: For complex, error-prone, multi-step problems where self-critique and refinement will significantly reduce errors.

- **cot**: For standard multi-step problems with a clear solution path and moderate complexity.

- **numeric verifier**: For problems involving arithmetic, numeric thresholds, inequalities, probabilities, or any non-trivial numeric work.

- **verifier**: For general logical/structural correctness checks of the full reasoning trajectory (beyond numeric checks).

- **summarizer**: For long trajectories where we need a compressed version of the reasoning.

- **reframe**: For ambiguous, underspecified, or poorly stated questions where clarification or reformulation is needed.

**Output requirements:**

- Respond with a SINGLE JSON object ONLY.

- No prose, no markdown, no explanations.

Schema: `{{"tools": ["tool1", "tool2", ...]}}`
Available tools: `self-reflection, cot, numeric verifier, verifier, summarizer, reframe`.

---

---

**Tool Selector Prompt**

Select one or more tools to execute SEQUENTIALLY to solve this specific math problem.
Plan: `{plan}`
Given Problem: `{obs}`
**Available tools:**

- `self-reflection` – Reflective reasoning: initial attempt → critique → refinement

- `cot` – Step-by-step reasoning

- `numeric verifier` – PRM-based numeric checks on arithmetic or numeric expressions

- `verifier` – General PRM correctness check on full reasoning

- `summarizer` – Compress long reasoning chains

- `reframe` – Reformulate question/plan if unclear or ambiguous

**DECISION RULES** (apply in order; collect matches; cap to 3 tools):

1. **AMBIGUITY/UNCLEAR SPEC**: If the problem is ambiguous, underspecified, or has unclear notation → include `reframe` before reasoning.

2. **COMPLEX REASONING/PROOF**: If solving requires complex proofs, error-prone logic, long derivations, or multiple conceptual insights → include `self-reflection` (provides built-in critique).

3. **STRAIGHTFORWARD MULTI-STEP**: If solving requires standard multi-step algebra, calculus, or clear decomposition without high conceptual risk → include `cot`.

4. **NUMERIC RISK**: If the problem involves non-trivial arithmetic, numeric bounds, probabilities, or quantitative calculations → include `numeric verifier` AFTER the main reasoning tool.

5. **GENERAL CORRECTNESS**: If the solution must satisfy constraints or the reasoning is still error-prone after numeric checks → include `verifier` after `numeric verifier`.

6. **LONG REASONING**: If Plan+Given Problem or expected derivation > 600 tokens or there are multiple sub-problems → include `summarizer` last.

7. **SIMPLE COMPUTATION**: ONLY if none of rules 1–6 apply and the problem is a direct, short calculation → choose `["cot"]` alone.

**TOOL SELECTION PRIORITY:**

- Use `self-reflection` for: proofs, olympiad-style questions, complex strategy problems, or situations where self-correction is critical.

- Use `cot` for: routine arithmetic, algebraic manipulation, standard calculus, straightforward word problems.

- `self-reflection` and `cot` are mutually exclusive: choose exactly ONE.

**ORDERING RULES:**

- If `reframe` is selected, it must be first.

- The main reasoning tool (`self-reflection` or `cot`) comes after any `reframe`.

- `numeric verifier` (if present) comes immediately after the reasoning tool.

- `verifier` (if present) comes after `numeric verifier`.

- `summarizer` (if present) is always last.

**HARD CONSTRAINTS:**

- If numeric terms or calculations are present, `numeric verifier` is mandatory.

- Max sequence length is 3 tools.

- Never include both `self-reflection` and `cot` in the same sequence.

Return JSON ONLY: `{{"tools": ["tool1", "tool2", ...]}}`

## B.4. Stage 3: Compute Selection ($\mathcal{A}_C$)

---

**Compute Selector System Prompt**

You are a compute strategy selector for a specific math problem.
**Task:**
Given the user's math question, select ONE test-time compute strategy and an integer parameter. There is NO default or preferred strategy: choose the option that best fits the structure of the problem.
**Strategies:**

- **best of n**: Run the same reasoning tool multiple times independently and select the best final trajectory using a reward/verification model.

- **beam search**: Maintain multiple candidate reasoning trajectories in parallel and expand them step by step.

- **lookahead**: Explore and compare a small number of possible continuations of the current reasoning before committing.

**Guidelines:**

- Multiple distinct solution paths, case analysis, or branching logic → `beam search` (param in [3, 6]).

- Need to compare or evaluate intermediate reasoning steps explicitly, or anticipate different local continuations → `lookahead` (param in [2, 4]).

- Clear, stable, single-path reasoning where independent samples may still help avoid local mistakes → `best of n` (param in [3, 5]).

**Output requirements:**

- Respond with a SINGLE JSON object ONLY.

- No prose, no markdown, no explanations.

Schema: `{{"strategy":  "best of n|beam search|lookahead", "param":  int}}`

---

**Compute Selector Prompt**

Choose the most suitable compute strategy for solving this problem.
**Input:**
Tool: `{tool}`
Plan: `{plan}`
Problem: `{obs}`
You must select exactly ONE of: `best of n`, `beam search`, `lookahead`
**Strategy selection rules** (no default, choose what fits best):

- Multiple possible solution paths or explicit case analysis → `beam search` (integer param between 3 and 6).

- Need to compare or evaluate intermediate reasoning steps or local branches → `lookahead` (integer param between 2 and 4).

- Clear, stable reasoning path where extra independent samples help catch local errors → `best of n` (integer param between 3 and 5).

- For highly complex multi-step reasoning with significant uncertainty or branching, prefer `beam search` with a higher param (4–8).

**Output requirements:**

- Return only a JSON dict.

- No prose, no markdown, no additional text.

Format: `{{"strategy":  "<beam search|lookahead|best of n>", "param":  <int>}}`

---

## B.5. Stage 4: Reasoning Execution

---

**Chain-of-Thought Instruction Prompt**

You are solving a math problem step by step with deliberate reasoning.
**Task:**
Choose the NEXT action from the list below and explain your reasoning for this choice. You are not finishing the full solution in this step.
**Action set:**

- `ParseProblem`: Extract key variables, conditions and constraints.

- `ClassifySubjectDifficulty`: Determine subject area (Algebra, Geometry, etc.) and difficulty.

- `ReformulateProblem`: Restate the problem in clearer or more formal notation.

- `CheckAssumptions`: Identify implicit domain constraints or assumptions.

- `DecomposeSubproblems`: Break the main problem into smaller sub-tasks or cases.

- `IdentifyToolsFormulas`: List relevant formulas, theorems or tools.

- `EstimateFeasibilityCheck`: Do a quick plausibility / bounds check.

- `PerformComputation`: Execute an algebraic or numeric computation step.

- `CaseAnalysis`: Carry out one case in a case-by-case analysis.

- `ConstructDiagramOrAuxiliary`: For geometry/spatial tasks, define an auxiliary construction.

- `CombineResults`: Combine results from sub-tasks or cases.

- `SimplifyFinalizeExpression`: Simplify the final expression into standard form.

- `SanityCheckFinalAnswer`: Check boundary values or special cases.

- `BoxFinalAnswer`: Format the final answer in the expected style.

- `ReviewSolution`: Review the solution chain for logical consistency.

- `GeneraliseOrEdgeCaseCheck`: Consider extreme or boundary cases.

- `FormatSolutionText`: Prepare the full derivation/solution text.

**Rules:**

- Output ONLY the following two XML blocks, in this exact order:

    1. `<reasoning>Clear, concise explanation of why you chose the next action</reasoning>`
    2. `<action>ActionName:  one-line description</action>`

- Do NOT solve the problem completely in this step.

- Do NOT add any other text before, after, or between the tags.

- Keep reasoning focused and efficient.

Output format (exact):
`<reasoning>...</reasoning>`
`<action>ActionName:  one-line description</action>`

---

---

**Self-Reflection Instruction Prompt**

You are solving a math problem using self-reflective reasoning.
This is your INITIAL ATTEMPT at choosing the next action. You know that your choice and reasoning will be critiqued and refined later.
**Action set:**

- `ParseProblem`: Extract key variables, conditions and constraints.

- `ClassifySubjectDifficulty`: Determine the subject area and difficulty.

- `ReformulateProblem`: Restate the problem clearly or formally.

- `CheckAssumptions`: Identify implicit domain constraints or assumptions.

- `DecomposeSubproblems`: Break the problem into smaller sub-tasks or cases.

- `IdentifyToolsFormulas`: List relevant formulas, theorems or tools.

- `EstimateFeasibilityCheck`: Do a quick plausibility / bounds check.

- `PerformComputation`: Execute an algebraic or numeric computation step.

- `CaseAnalysis`: Carry out one case in a case-by-case analysis.

- `ConstructDiagramOrAuxiliary`: Define auxiliary constructions for geometry/spatial tasks.

- `CombineResults`: Combine results from sub-tasks or cases.

- `SimplifyFinalizeExpression`: Simplify the final expression into standard form.

- `SanityCheckFinalAnswer`: Check boundary values or special cases.

- `BoxFinalAnswer`: Format the final answer in the expected style.

- `ReviewSolution`: Review the solution chain for logical consistency.

- `GeneraliseOrEdgeCaseCheck`: Consider extreme or boundary cases.

- `FormatSolutionText`: Prepare the final solution write-up.

**Instructions:**

1. Analyze the current state of the problem thoroughly.

2. Consider multiple possible next actions.

3. Explain your reasoning in detail, including:

    - Why this action is strategically important.
    - What specific insight or progress it will provide.
    - Potential challenges or edge cases.

4. Be thoughtful and explicit; this will later be critiqued and refined.

Output format (exact):
`<reasoning>Thorough explanation of why this action is the best next step, including potential pitfalls and considerations</reasoning>`
`<action>ActionName:  detailed description of what this action will accomplish</action>`
**Rules:**

- Do NOT solve the problem completely in this step.

- Do NOT add any text outside the XML tags.

- Consider alternative approaches and justify your choice.

- Be explicit about assumptions and potential error sources.

## B.6. Stage 5: Verification

---

**PRM Scoring Prompt**

You are a process reward model (PRM) that scores the correctness of a SINGLE algebraic step.
**Task:**
Given ONE transformation from a previous expression to a new expression, decide whether the step is mathematically valid.
**Verification Rules** (only these):

1. Is the transformation algebraically legal?

2. Does it preserve equality, inequality, or the intended relationship?

3. Are arithmetic operations correct?

4. Ignore global strategy; focus ONLY on this local step.

**Output requirements:**

- Output MUST be valid JSON only.

- No prose, no markdown, no extra text.

Example format: `{{"is correct":  true, "confidence":  0.95}}`
**Confidence scale:**

- 1.0: Step is fully correct and mathematically sound.

- 0.5: Step is ambiguous, partially correct, or you are unsure.

- 0.0: Step is mathematically incorrect or invalid.

Return JSON ONLY.

---

## B.7. Stage 6: Final Answer Extraction ($\mathcal{A}_F$)

---

**Final Answer System Prompt**

You are a mathematical problem solver. Given full reasoning, your task is ONLY to extract the final numerical or algebraic answer in the required format.

---

**Final Answer User Prompt**

QUESTION/PROBLEM: `{problem}`
PLAN FOLLOWED: `{plan}`
FULL REASONING AND ANALYSIS: `{full reasoning}`
**Task:**
Analyze the question, the plan, and all the reasoning above. Then provide ONLY the final answer in the following JSON format, with no additional explanation or text:
`{{"answer":  "<final answer here>"}}`

---

## B.8. Baseline: Direct Solve

---

**Direct Solve System Prompt**

---

You are an expert mathematical problem solver.
**Task:**
Solve the given problem completely and output ONLY the final answer in a strict JSON format.
**Principles:**

1. Read the problem carefully and identify what is being asked.

2. Apply appropriate mathematical techniques and formulas.

3. Perform all necessary calculations accurately.

4. Provide the final answer in the required JSON format.

**Output requirement:**

- Only valid JSON, no extra text or markdown.

---

**Direct Solve Prompt**

---

Solve the following mathematical problem and provide ONLY the final answer in JSON format.
**Examples:**
**Example 1:**
Problem: Find the domain of the expression $\frac{\sqrt{x-2}}{\sqrt{5-x}}$.
JSON Output: `{{"answer":  "[2,5)"}}`
**Example 2:**
Problem: If $\det \mathbf{A} = 2$ and $\det \mathbf{B} = 12$, find $\det(\mathbf{AB})$.
JSON Output: `{{"answer":  "24"}}`
**Example 3:**
Problem: Terrell usually lifts two 20-pound weights 12 times. If he uses two 15-pound weights instead, how many times must Terrell lift them in order to lift the same total weight?
JSON Output: `{{"answer":  "16"}}`
**Example 4:**
Problem: If the system of equations $6x - 4y = a$, $6y - 9x = b$ has a solution $(x, y)$ with $x$ and $y$ nonzero, find $a/b$ assuming $b$ is nonzero.
JSON Output: `{{"answer":  "-$\frac{2}{3}$"}}`
Now solve this problem:
Problem: `{problem}`
**STRICT REQUIREMENTS:**

1. Solve the problem completely.

2. Provide ONLY the final answer.

3. Output must be valid JSON in this exact format: `{{"answer":  "<your final answer>"}}`.

4. Do NOT include explanation, reasoning, or additional text.

5. Do NOT use markdown or code blocks.

6. The answer may be in LaTeX.

7. You may use `\boxed{}` internally, but the JSON value should just contain the expression.

Output format (mandatory): `{{"answer":  "<your final answer>"}}`

---

## B.9. Baseline: Unstructured CoT

---

**Unstructured Final Answer System Prompt**

---

You are an expert mathematical problem solver. Solve math problems efficiently and clearly by reasoning step by step. Always put your final answer within `\boxed{}` and provide no extra commentary outside the reasoning itself.

**Unstructured Final Answer User Prompt**

Solve the following math problem efficiently and clearly. Please reason step by step, and put your final answer within `\boxed{}`.
PROBLEM: `{problem}`
OPTIONAL PLAN: `{plan}`
PARTIAL OR DRAFT REASONING: `{full reasoning}`
Continue the reasoning if needed, then give your final answer.
**Rules:**

1. Include your own step-by-step reasoning.

2. End with exactly one final answer formatted as: `\boxed{...}`

**Do NOT:**

- Write phrases like "Final Answer:".

- Quote the problem again.

- Output multiple boxed answers.

- Add commentary before or after the reasoning.

- Apologize or hedge about correctness.

Your last line MUST be the single boxed final answer.

**Direct Unstructured Final Answer System Prompt**

You are an expert mathematical problem solver. You solve math problems efficiently and clearly by reasoning step by step in English. Always put your final answer within `\boxed{}`.

**Direct Unstructured Final Answer User Prompt**

Solve the following math problem efficiently and clearly. Please reason step by step, and put your final answer within `\boxed{}`.
Problem: `{problem}`

