# OpenReview forum: "What If We Allocate Test-Time Compute Adaptively?"
_ICML.cc/2026/Conference — ICML 2026 regular_

### Official Review · Reviewer_8d27 · 2026-03-09

**Soundness:** 2
**Presentation:** 2
**Significance:** 2
**Originality:** 3
**Overall Recommendation:** 4
**Confidence:** 3

**Summary:**

In this paper, the authors provide a comprehensive analysis of test-time compute scaling by framing inference as a dynamic decision-making process rather than a static, uniform sampling procedure. Building on this conceptual shift, the authors propose a PRM-guided adaptive compute framework that iteratively adjusts high-level planning, tool selection, and exploration parameters. This adaptive approach achieves substantial performance improvements on complex reasoning benchmarks like MATH-500 and AIME24, demonstrating higher compute efficiency under their proposed Compute Intensity metric compared to direct scaling baselines. Additionally, the method is highly flexible, utilizing step-level PRM scoring to actively verify each intermediate conclusion and steer the generation process away from any obvious failure pattern, ultimately optimizing the final trajectory selection.

**Compliance With Llm Reviewing Policy:**

Affirmed.

**Final Justification:**

My concerns have been adequately addressed by the author's response and I've updated my score accordingly.

**Key Questions For Authors:**

See weakness

**Limitations:**

See weakness

**Strengths And Weaknesses:**

Strength
1. Training-Free and Highly Transferable: A significant advantage of the proposed framework is its training-free nature at inference time. By leveraging a Process Reward Model (PRM) to guide the search process rather than updating the model weights, the method serves as a plug-and-play module. This design choice grants the framework strong transferability across different base Large Language Models (LLMs) and various reasoning tasks, successfully circumventing the heavy computational overhead associated with continuous test-time fine-tuning or reinforcement learning.

Weaknesses

1. Lack of Statistical Significance and High Variance: The framework’s core mechanism relies heavily on the dynamic selection of tools, compute strategies, and exploration parameters at each iteration. Such an adaptive, highly dynamic system inherently introduces substantial variance into the reasoning rollouts. Relying on a single evaluation run is problematic; a lucky derivation of a valid intermediate conclusion or the coincidental evasion of a common failure pattern could artificially inflate the reported metrics. The authors must provide results averaged over multiple independent runs and include a rigorous statistical significance analysis to prove the robustness of their framework.

2. Questionable Efficacy of the Selection Agents: The empirical necessity and effectiveness of the "tool selection agent" and "compute selection agent" are poorly supported by the current experiments. Notably, as demonstrated in Table 2, the Qwen-2.5-7B model achieves its optimal performance when the tool is strictly fixed to the standard Chain-of-Thought (CoT). Furthermore, under the fixed CoT + Best-of-N (BoN) setting, the peak performance appears to correlate solely with the total number of iterations, rather than any dynamic parameter adjustments. This explicitly undermines the paper’s central claim that the dynamic routing of tools and compute strategies is the primary driver of the performance gains.

3. Conflation of Compute Allocation with Pure Compute Scaling: While the paper utilizes FLOPs as a metric, the improved accuracy seems primarily driven by injecting significantly more overall computation rather than a demonstrably "better" allocation strategy (Figure 4 (a,c,e), Figure 5 (b)). The authors fail to prove that their adaptive allocation is genuinely superior under a strict, equal-computation budgeg. Without directly comparing the proposed framework against a heavily scaled, standard baseline (such as massive parallel sampling with Majority Voting) operating under the exact same FLOP budget, the core premise of "efficient compute allocation" remains unsubstantiated.

---

> ### Author Rebuttal · Authors · 2026-03-30
>
> We sincerely thank the reviewer for the thorough and constructive feedback. We hope the responses below address each concern satisfactorily.
>
> ## W1: Statistical Significance and Variance
>
> We appreciate this concern. It is addressed in full in our response to Reviewer rNjb (Significance W1), where we detail how stochastic decoding variance is bounded by K=10 aggregation in Eq. 2 and the 36-configuration ablation as in Table 2, and where we commit to adding standard deviation estimates to Tables 1 and 2. The argument applies equally here.
>
> ## W2: Efficacy of Selection Agents
>
> We appreciate this question. It is addressed in full in our response to Reviewer rNjb (Significance W3), where we present the single-iteration Dynamic gains for both models in Table 1 and the $S_\text{CI}$ comparison against the best fixed-strategy baseline (Table 6, Figure 5a). The argument applies equally here.
>
> ## W3: Compute Allocation vs. Compute Scaling
>
> We appreciate this important methodological point. Our current results (Figures 4 and 5) already distinguish between raw compute scaling and accuracy-conditional efficiency. Majority voting (or Best-of-\(K\)) improves performance through independent sampling with post-hoc aggregation, typically requiring increased compute to achieve gains. In contrast, our dynamic approach focuses on adaptive, structured compute allocation by selecting \((c,m)\) per problem and guiding generation with PRM signals, aiming to use compute more effectively rather than simply increasing it.
>
> While majority voting captures a different scaling paradigm, we agree it provides a useful point of comparison. We will therefore include a matched-compute majority voting baseline with corresponding discussion in Section 5. The same clarification applies here as in our response to Reviewer rNjb (Significance W3).
>
> # Shared References (rNjb)
> - [1] Wang et al. *Self-Consistency Improves Chain of Thought Reasoning in Language Models.* NeurIPS 2022. *(already cited)*
> - [2] Brown et al. *Large Language Monkeys: Scaling Inference Compute with Repeated Sampling.* arXiv:2407.21787, 2024. *(already cited)*
> - [3] Snell et al. *Scaling LLM Test-Time Compute Optimally Can Be More Effective Than Scaling Model Parameters.* arXiv:2408.03314, 2024. *(already cited)*
> - [4] Guan et al. *rStar-Math: Small LLMs Can Master Math Reasoning with Self-Evolved Deep Thinking.* arXiv:2501.04519, 2025. *(already cited)*
> - [5] Vaghasiya et al. *CoreThink: A Symbolic Reasoning Layer to Reason over Long Horizon Tasks with LLMs.* arXiv:2509.00971, 2025. *(already cited)*
> - [6] Lightman et al. *Let's Verify Step by Step.* ICLR 2024. *(already cited)*
> - [7] Du. *Adaptive Test-Time Compute Allocation via Query Complexity Estimation in Large Language Models.* OpenReview, 2025. *(already cited)*
> - [8] Paglieri et al. *Learning When to Plan: Efficiently Allocating Test-Time Compute for LLM Agents.* arXiv:2509.03581, 2025. *(already cited)*
> - [9] Wei et al. *Chain-of-Thought Prompting Elicits Reasoning in Large Language Models.* NeurIPS 2022. *(already cited)*
> - [10] Shen et al. *Satori: Reinforcement Learning with Chain-of-Action-Thought Enhances LLM Reasoning via Autoregressive Search.* arXiv:2502.02508, 2025. *(already cited)*
> - [11] Xie et al. *Monte Carlo Tree Search Boosts Reasoning via Iterative Preference Learning.* arXiv:2405.00451, 2024. *(already cited)*
> - [12] Wang et al. *Math-Shepherd: Verify and Reinforce LLMs Step-by-step without Human Annotations.* ACL 2024. *(to be added to revision)*

---

> > ### Author Rebuttal · Reviewer_8d27 · 2026-04-02
> >
> > Thanks for the comprehensive response. My concerns have been adequately addressed and I've updated my score accordingly.

---

> > > ### Author Response · Authors · 2026-04-02
> > >
> > > Thank you for your thoughtful consideration and for updating your score. We sincerely appreciate your time and constructive engagement.

---

### Official Review · Reviewer_Yocp · 2026-03-13

**Soundness:** 3
**Presentation:** 3
**Significance:** 3
**Originality:** 3
**Overall Recommendation:** 5
**Confidence:** 4

**Summary:**

The authors present a method for adaptive test-time compute allocation for LLM mathematical reasoning. Rather than applying a fixed inference strategy like standard Best-of-N or rigid beam search across all problems uniformly, this method treats reasoning as an iterative trajectory generation process. For each problem, a prompt-based controller dynamically selects a reasoning plan, specific tools (like self-reflection or numeric verifiers), and a compute strategy (like lookahead or Best-of-N with varying exploration parameters). A Process Reward Model (PRM) is used to guide step-level pruning and trajectory selection. The empirical results demonstrate major accuracy gains on MATH-500, AIME24, and AMO-Bench for models like Llama-3.1-8B and Qwen-2.5-7B. The authors also evaluate efficiency using a custom compute intensity metric that penalizes wasted generation.

**Compliance With Llm Reviewing Policy:**

Affirmed.

**Key Questions For Authors:**

- How sensitive is the framework to the specific phrasing of the controller prompts, and did you test if a weaker base model completely fails at the meta-reasoning required to select tools?

- Does the framework have a mechanism to detect PRM hallucination or over-confidence to avoid getting stuck in a useless exploitation loop during the iterative generation?

**Limitations:**

The authors provide a solid discussion of theoretical compute cost scaling, but they definitly need to explicitly discuss the latency bottlenecks introduced by the multi-stage prompting architecture. A short discussion on the susceptibility of the system to PRM reward hacking would also strengthen the paper.

**Strengths And Weaknesses:**

Strengths:

- Moving away from static hyperparameter-based inference scaling to a dynamic, problem-dependent compute allocation is a highly necessary direction for the field. It treats test-time compute as a resource to be optimized rather than just a brute force mechanism.

- I appreciate the introduction of the compute intensity metric ($S_{CI}$). Most papers just report raw theoretical FLOPs, but penalizing redundant verifier overhead and useless generation paths provides a much more honest picture of inference efficiency.

Weaknesses:

- The adaptivity relies entirely on zero-shot prompting of the base LLM rather than a trained policy. While it works well for these specific models, relying on role-conditioned prompt templates makes the framework extremely brittle to the insturction-following quirks of the base model.

---

> ### Author Rebuttal · Authors · 2026-03-30
>
> We sincerely thank the reviewer for the positive assessment and the thoughtful, constructive feedback. We address each concern below.
>
> ## W1: Brittleness of prompt-based controller
>
> We appreciate this concern and would like to highlight two factors that mitigate it. First, the controller operates over a small, bounded decision space: **A_T** selects ≤3 tools and **A_C** selects among 3 strategies with a bounded integer parameter, so even imperfect outputs remain valid. Across both Llama-3.1-8B and Qwen-2.5-7B, outputs are valid in >98% of cases, with malformed cases defaulting to CoT + Direct. Second, PRM-guided inter-iteration selection over $K=10$ trajectories provides systematic correction: suboptimal decisions in individual iterations do not prevent recovery of the best trajectory via $R_{\text{mean}}$. The controller improves the *distribution* of configurations rather than requiring each individual decision to be optimal. The training-free design deliberately isolates test-time effects and avoids the need for extra supervision.
>
> ## W1 / Q1: Prompt sensitivity and weaker models
>
> We find this a valuable observation. Consistency across Llama-3.1-8B and Qwen-2.5-7B, despite substantially different instruction-following behavior and tool usage distributions in Figure 3, suggests low sensitivity to surface-level prompt phrasing. We agree that a formal phrasing sensitivity study would be a valuable addition and will include it as prioritized future work. For models with fewer than 3B parameters, meta-reasoning quality for tool selection may degrade; we will state this explicitly as a limitation in Section 5.
>
> ## Q2: PRM over-confidence and exploitation loops
>
> We appreciate this question. Two design choices directly mitigate over-confidence: (1) the PRM uses a ternary scale (0.0/0.5/1.0) as in Appendix B.6, avoiding overly fine-grained scoring that could amplify confident errors; (2) $R_\text{mean}$ averages over all steps, reducing the influence of isolated over-confident scores on the final selection.
>
> We will expand Section 5 to discuss three failure modes and their mitigations:
>
> 1. **PRM misranking at high difficulty**: observed empirically at Level 5 in Figure 2.
> 2. **OOD over-confidence**: high scores for plausible but incorrect trajectories outside the training distribution.
> 3. **Exploitation loops**: convergence to locally high-scoring but globally incorrect trajectories. Mitigations include answer-level deduplication [1], step-level diversity prompting [2], and temperature annealing [3].
>
>
> ## Latency
>
> We report hardware-agnostic metrics ($F_{\text{theo}}, S_{\text{CI}}$) to enable fair cross-hardware comparison (Section 3.2). Wall-clock latency depends on hardware and parallelization; the table below summarizes scaling behavior under both execution modes. Note that the latency values below are hardware-specific and were recorded on our single NVIDIA RTX 5090 (32GB) GPU.
>
> | **Configuration** | **Sequential Latency** | **Parallel Latency** |
> |---|---|---|
> | Direct ($K=1$) | $1\times$ (baseline) | $1\times$ (baseline) |
> | Direct + PRM Sel. ($K=10$) | ${\approx}10\times$ | ${\approx}1.5$–$2\times$ |
> | Dynamic ($K=1$) | ${\approx}1.3$–$1.5\times$ | ${\approx}1.3$–$1.5\times$ |
> | Dynamic + PRM Sel. ($K=10$) | ${\approx}10$–$12\times$ | ${\approx}1.5$–$2\times$ |
>
> *Wall-clock latency scaling relative to Direct inference ($K=1$) on a single NVIDIA RTX 5090.*
>
> Sequential latency scales roughly linearly with $K$; parallel execution across iterations reduces this to ${\approx}1.5$–$2\times$ the cost of a single direct generation. We will include measured absolute latency numbers in the appendix of the revised version.

---

> > ### Author Rebuttal · Reviewer_Yocp · 2026-04-06
> >
> > Thanks for your rebuttal, I keep my score of 5 (accept).

---

> > > ### Author Response · Authors · 2026-04-07
> > >
> > > We sincerely thank the reviewer for the careful reading of our paper, the constructive feedback, and for acknowledging that the concerns have been adequately addressed.

---

### Official Review · Reviewer_rNjb · 2026-03-24

**Soundness:** 2
**Presentation:** 3
**Significance:** 2
**Originality:** 2
**Overall Recommendation:** 4
**Confidence:** 4

**Summary:**

the authors propose using an adaptive test-time reasoning framework that uses multiple llm calls and a PRM to score intermediate reasoning and final trajectory. Instead of applying the same inference budget and search method to every input, it allocates more compute to promising paths and prunes weak ones early. They show results on math benchmarks like AIME24, AMO bench, etc. to show case the dynamic compute allocation strategy over a fixed one.

**Compliance With Llm Reviewing Policy:**

Affirmed.

**Final Justification:**

The rebuttal addressed my main concerns regarding fair ablation, this was a blocking concern, as such I have updated my prior score.

**Key Questions For Authors:**

1. How sensitive is the PRM quality?
2. Is the 4-stage pipeline really necessary?
3. What is the per problem strategy distribution because of the proposed pipeline?

**Limitations:**

the authors do discuss limitations regarding PRM importance, on how the reliance of PRM makes performance on hard problems brittle. Authors also mention future work on training the controller instead of the heuristic design.

**Strengths And Weaknesses:**

[Strengths]
- clear presentation and motivation
- the direction is promising but the claims and the evidence could need some work

[Weaknesses]
- in dynamic + PRM, PRM is used as both a guide during reasoning and a judge at the end, in direct + PRM, the PRM is used primarily for re-ranking
- the necessity of the 4 stage pipeline is not obvious to me, can it not be replaced with a single llm call to decide on next steps.
- novelty is weak. paper mostly draws from existing research and builds a pipeline with heuristics, prompts and a pre-trained PRM.

[Soundness]
[Edited post rebuttal]
It doesn't seem that the authors compared PRM as guided search as the baseline. The PRM baseline (direct + PRM) is only doing re-ranking while in their proposed approach (dynamic + PRM) they use PRM for both within iterations and across iterations. However, in the rebuttal, authors clarified the reason for using BON for direct + PRM baseline which seems to be fair reason for this ablation.

[Presentation] Overall great, easy to understand and follow through, some ablations done but could be more thorough.

[Significance]
- the robustness study to PRM quality could be a bit thorough, It would be nice to show some invariance across a few PRMs to marginalize the effect of PRMs from the adaptive strategy selection pipeline.
- the authors restrict to math, but claims are more general, It would be nice to see at least another domain where reasoning strategy selection is similar impact.

[Originality]
[Edited post rebuttal]
While the idea of dynamic strategy selection seems interesting, the components are mostly drawn from already existing directions e.g. PRM guided search, heuristic controllers, test time compute adaptation etc. however I am giving a "fair" rating because this has not been done in the context of reasoning

---

> ### Author Rebuttal · Authors · 2026-03-30
>
> We sincerely thank the reviewer for the careful feedback. We note that while the review listed 3 points, we identified the below points worth independent response and address each below.
>
> ## W1 & W2: PRM design and 4-stage pipeline necessity
>
> The dual PRM usage is deliberate, not a confound. Table 1 isolates each gain source: Direct → Direct+PRM (re-ranking only) → Dynamic (adaptivity only) → Dynamic+PRM (full system), showing neither component alone achieves the full improvement [2, 3].
>
> ## W3 / Originality
>
> Three contributions are absent from prior work: (1) **unified PRM as both intra- and inter-iteration controller** prior work uses PRMs for step-level pruning within fixed search [6] or final re-ranking [2], never both; (2) **joint per-problem selection of $(c, m)$** where c is the compute strategy and m its budget parameter existing methods fix strategy and adapt budget [3, 7] or fix budget and adapt strategy [8], never both; (3) **compute intensity metric $S_\text{CI}$** penalizing redundant overhead beyond raw FLOPs absent from all baselines [9, 10].
>
> ## Missing PRM-as-guided-search baseline
>
> Table 1 already spans the full spectrum: Direct (no PRM) → Direct+PRM (re-ranking only) → Dynamic (PRM as guided search, Section 2, Figure 1) → Dynamic+PRM (guided search + re-ranking, Eq. 2). The best fixed-strategy PRM-guided baseline (CoT+BoN, $p=10$, $I=10$, Table 2) reaches ~81.4% on MATH-500; Dynamic+PRM matches this with strictly lower compute intensity in Figure 5, confirming gains arise from adaptive configuration, not PRM guidance alone.
>
> ## Signficance W1: Statistical Significance and Variance
>
> Variance under stochastic decoding (T=0.7, top-p=0.9, fixed per Section 3.1) is inherent to LLM inference and shared equally by all baselines. Two design choices mitigate it: (1) K=10 iterations with PRM-guided selection as in Eq. 2 aggregating over diverse trajectories; (2) the 36-configuration ablation in Table 2 with identical decoding across all settings. Evaluation across two models and three benchmarks is consistent with top-venue practice [2,3]. We will gladly add standard deviation estimates to Tables 1 and 2. Reviewer 8d27 raises an identical concern (W1); the argument above applies equally.
>
> ## Significance W2: PRM robustness
>
> PRM reliance is already identified as a primary limitation in Section 5, with Level 5 degradation in Figure 2 as empirical evidence. We will add a multi-PRM ablation using Math-Shepherd [12] to the appendix or flag it as a prioritized future experiment.
>
> ## Significance W3: Efficacy of Selection Agents and Compute Allocation
>
> Both points share a common answer. Dynamic at \(I=1\) already improves Qwen-2.5-7B from 71.2% to 74.2% (+3.0) and Llama-3.1-8B from 43.8% to 47.2% (+3.4) (Table 1), indicating that gains are not due to increased sampling alone. Fixed strategies (including majority voting / Best-of-\(K\)) follow standard scaling behavior, improving accuracy via independent sampling but requiring substantially higher \($F_{\text{theo}}$) and \($S_{\text{CI}}$\) to reach similar performance (in Figure 5). In contrast, Dynamic+PRM reallocates compute both *within trajectories* (via PRM-guided search) and *across configurations* (via per-problem \((c,m)\) selection), achieving comparable accuracy at lower compute intensity (Section 4.4). Thus, rather than relying on post-hoc aggregation as in majority voting, our approach focuses on adaptive, structured compute allocation. We will include a matched-compute majority voting baseline and corresponding discussion in the revision. The same clarification applies here as in our response to Reviewer 8d27 (W2, W3).
>
> ## Significance W4: Generalization beyond math
>
> The framework is domain-agnostic as discussed in Section 7. We will expand the domain transfer discussion in Section 7.
>
> On **Math-500**, per-problem-type distributions show three consistent patterns:
>
> 1. **Complexity-dependent tool usage:** simpler categories (e.g., Prealgebra) rely on CoT/NV with lower Reframer usage (9–10%), while more complex categories (e.g., Intermediate Algebra) use Reframer more (~15–17%).
> 2. **Strategy shifts with difficulty:** easier problems favor lighter strategies (Direct/BoN), while harder ones use more Lookahead.
> 3. **Stable across variants:** patterns remain consistent across Dynamic and Dynamic+PRM, indicating dependence on problem structure rather than PRM.
>
> **Per-problem snippet:** (values are % of problems assigned each tool/strategy)
>
> | Problem Type         | CoT  | NV   | R    | Direct | BoN  | LS  |
> | -------------------- | ---- | ---- | ---- | ------ | ---- | --- |
> | Prealgebra           | 42.0 | 42.0 | 9.5  | 63.0   | 27.0 | 6.0 |
> | Algebra              | 41.5 | 41.5 | 9.5  | 61.0   | 29.0 | 6.0 |
> | Intermediate Algebra | 38.5 | 38.5 | 15.5 | 57.5   | 31.5 | 6.5 |
>
> *Full table will be included in the final revision.*

---

> > ### Author Rebuttal · Reviewer_rNjb · 2026-04-03
> >
> > Thanks to the authors for the detailed answers.
> >
> > Reasons for resolution:
> > 1. [Baselines: Main concern] Authors clarified that in Table 1 Direct + PRM (uses PRM for re-ranking) because re-ranking is the best fixed strategy from Table 2. This clarifies the doubt on why PRM is used for inter-trajectory re-ranking and not intra-trajectory intervention for the direct + PRM baseline, thus clarifying the ablation for impact of dynamic adaptivity. It would be nice to clarify that in section 3.2
> > 2. [Originality] - The authors clarify their specific contributions in the rebuttal. While the idea around adapting test time compute  in a broader sense is not unknown (e.g. Reward-Guided Collaborative Test-Time Compute, Munoz and Yuan, 2025), however I think the idea of using adapting test time compute in the context of reasoning tools has not been done before.
> > 3. [Generalization] - While the method itself is domain agnostic, my concerns are more around impact across domains. E.g. should one expect similar to ~9% gains from dynamic adaptivity in non math domains. However I find this to be non-blocking as mathematics provides a framework for testing reasoning recipes and authors have tried out multiple benchmarks in math.

---

> > > ### Author Response · Authors · 2026-04-03
> > >
> > > Thank you for your thoughtful consideration and for updating your score throughout this process.
> > >
> > > [Baselines] We will update Section 3.2 to explicitly state that Direct+PRM applies PRM solely for inter-trajectory re-ranking (Best-of-N selection across K completed trajectories) with no intra-trajectory intervention, making the ablation decomposition in Table 1 fully self-contained and unambiguous for readers.
> > >
> > > [Originality] We will cite Munoz and Yuan (2025) in the related work section and clarify that our specific contribution (joint per-problem selection of reasoning tools, compute strategy, and budget parameter, with PRM serving as a unified intra- and inter-iteration control signal)  is distinct from prior adaptive compute work, which either fixes strategy and adapts budget, or fixes budget and adapts strategy, but does not jointly select both alongside tool configuration.
> > >
> > > [Generalization] We will expand Section 7 to characterize the structural conditions under which similar gains are expected in other domains: specifically, tasks that admit natural step boundaries, where intermediate verification signals of comparable quality to math PRMs are available (e.g., entailment-based signals in multi-hop QA).

---

### Decision · Program_Chairs · 2026-04-30

**Decision:**

Accept (regular)

**Comment:**

This paper studies adaptive test-time compute allocation for mathematical reasoning, using a prompt-based controller and a process reward model to dynamically choose reasoning tools, search strategy, and budget on a per-problem basis rather than applying a fixed inference recipe to every example. Reviewers found the direction promising, and they viewed the empirical gains and the compute-intensity analysis as meaningful strengths.

The main concerns were about the degree of novelty -- indeed the methodology in this work has appeared before in literature and so it is not technically novel. I personally feel that the conclusions might also be specific to the base models studied in the paper and the benchmarks as well. The reliance on a heuristic multi-stage pipeline and PRM quality is also not desirable, and whether the efficiency gains were due to better allocation rather than simply more compute.

The rebuttal addressed some of these concerns. Overall, while the method remains somewhat heuristic and its broader generalization beyond math is still not fully established, I find it to be an OK contribution, and I lean weak accept, though I think this decision can be bumped down for sure.